# Visualizing Thought: Conceptual Diagrams Enable Robust Planning in LMMs

## Abstract

Human reasoning relies on constructing and manipulating mental models—simplified internal representations of situations used to understand and solve problems. Conceptual diagrams (e.g., a sketch drawn to aid reasoning) externalize these mental models, abstracting irrelevant visual details to efficiently capture how entities interact. In contrast, Large Language Models (LLMs) and Large MultiModal Models (LMMs) predominantly reason through text, limiting their effectiveness on complex multi-step tasks. In this paper, we propose Visual Thinking, a generalizable framework that enables LMMs to reason through multiple chains of self-generated conceptual diagrams, significantly enhancing their combinatorial planning capabilities. Our approach requires no human input beyond the natural language description of the task. It integrates textual and diagrammatic reasoning within an optimized Graph-of-Thought inference framework, enhanced by beam search and depth-wise backtracking. Evaluated on multiple challenging PDDL planning domains, our method substantially improves LMM performance (e.g., GPT-4o: $35.5\% \rightarrow 90.2\%$ in Blocksworld) and consistently outperforms text-only search-based inference methods. Additionally, on more difficult planning domains with solution depths up to 40, our approach outperforms the o1-preview reasoning model (e.g., 16 percentage points improvement in Floor Tiles). These results highlight the value of conceptual diagrams as a reasoning medium in LMMs.

## 1 Introduction

Natural language is a powerful medium for communication, enabling humans to effectively share knowledge and ideas [47, 11, 53]. However, language alone is not an optimal medium for reasoning, as it is inherently linear, sequential, and verbose, making it inefficient for representing complex logical and relational structures [34, 26, 54]. Prior evidence 'human thought' is inherently not verbal, sequential, or linear; rather, it is spatial, parallel, and image-like [54]. Humans construct and utilize internal mental models—simplified analogues of real or hypothetical situations [32, 21, 26, 6], and dynamically manipulate them to represent and predict interactions between objects and solve problems. Crucially, mental models are multimodal, integrating both visual and verbal representations to facilitate learning and robust reasoning [39]. Finally, visual representations have always played a central role in human reasoning and communication, from prehistoric cave art, which predates written language [12], to modern textbook diagrams, scientific figures, and blackboard sketches.

Conceptual diagrams are simplified visual representations that use basic shapes (e.g., circles, squares, lines) to capture how entities interact while abstracting away irrelevant details [54, 34]. They externalize internal mental models, reducing cognitive load and enabling rapid perceptual inference and clearer reasoning [26, 21]. Unlike photorealistic images, which capture fine-grained details of how objects appear, conceptual diagrams encode the structural and relational information essential for reasoning, using colors, relative positions and sizes, and annotations [54, 34]. For example, a square in a diagram might represent a complex object such as a car, with its color, relative size, and position visually encoding relationships to other entities while omitting irrelevant appearance details. Thus, conceptual diagrams are an effective reasoning medium complementary to language, overcoming language's limitations in representing relational structure and aligning closely with humans' multimodal reasoning [54, 26, 20].

Modern large language models (LLMs) and large multimodal models (LMMs) [42, 43, 4] have achieved remarkable success on mathematical and scientific benchmarks, including GSM8K [13], MATH [28], and GPQA [70]. Despite these advances, their reasoning remains inconsistent, particularly on multi-step compositional reasoning, long-horizon planning, and tasks requiring backtracking or error correction [20, 57, 15, 9]. These limitations stem partly from LLMs' reliance on language, which is inherently linear and inefficient for representing complex relational structures [20, 57, 8]. Moreover, the autoregressive architecture of current models enforces sequential next-token prediction, making

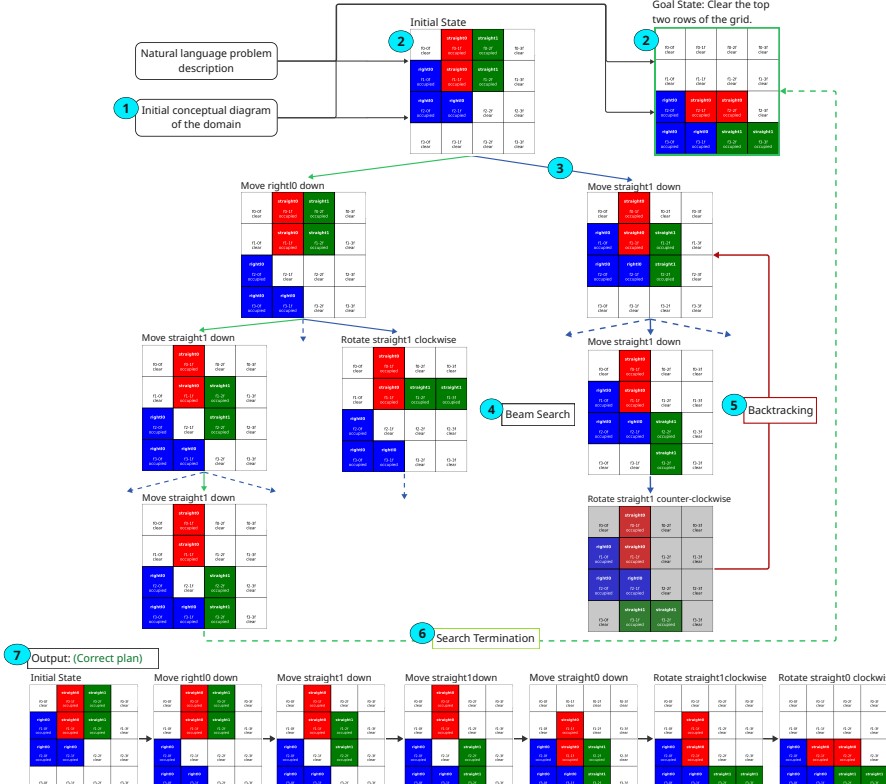

Figure 1: **Our proposed approach**. Example diagrams are from the Tetris domain, where tiles are moved on a grid to reach a goal state. (1) The model generates multiple diagram schemas and codes for a random instance; their rendered diagrams are ranked, and the code of the top choice is cached (Fig. 2). (2) Conditioned on this code, diagrams for the initial and goal states are generated (Fig. 3). (4) Beam search ranks all candidates at a depth by proximity to the goal, expanding the top $k = 4$ (Sec. 3). (5) Depth-wise backtracking is applied when all candidate states at a depth fail validation, returning to the deepest available ancestor. (6) The process stops when the goal is reached or a maximum number of steps is visited. (7) The output is the action sequence (plan) plus textual and diagrammatic representations of intermediate states.[1]

backtracking challenging [20]. Thus, enabling LMMs to reason with conceptual diagrams and backtrack within a graph-based inference framework offers a promising approach to overcome these bottlenecks.

In this work, we propose Visualizing Thought, a framework that enables LMMs to solve combinatorial problems through multiple multimodal chains of self-generated conceptual diagrams and textual reasoning. Our approach requires no domain-specific modifications or manual engineering to solve any combinatorial problem expressible in the Planning Domain Definition Language (PDDL) [40], given only a natural language specification of the initial state, goal state, and possible actions. Importantly, our method does not rely on predefined visual templates or geometric priors; instead, it generates conceptual diagrams directly from the textual problem description.

Visualizing Thought decomposes inference into a graph of intermediate reasoning steps, where at each node the model selects the next best state. Each node is multimodal, containing both a textual description of the state and a corresponding conceptual diagram (see Fig. 1). At each step, the LMM (i) generates the next state conditioned on the textual and diagrammatic representations of the states in the action path; (ii) produces a diagram schema—a structured set of statements specifying each object's shape, relative size and location, and status; and (iii) generates Matplotlib code from the schema that renders the state diagram. To ensure inference quality, we incorporate guardrails such as diagram-schema self-reflection checks and local (parent–child) and global (entire path from the initial state) validity checks. To manage the exponential growth of the combinatorial search space [7, 68], we integrate beam search to rank validated candidate states at each inference depth and expand only the top $k$. We also incorporate depth-wise backtracking, which allows the model to revisit earlier validated nodes if all current candidates fail verification. Together, these components enable more efficient exploration of the search space.

---

[1] For Figs. 1 and 4, we adjusted the diagram codes only to increase font sizes for better readability. Figs. 2-3 are unmodified generations. Fig. 1 actions are simplified; full action strings include previous/current cells occupied.

| Domain | Single inference | | | Search-based inference (base LLM = GPT-4o) | | | |
|---|---|---|---|---|---|---|---|
| | GPT-4o | o1-mini | o1-preview | GoT | Optimized GoT | RAP[20] | Visual Thinking |
| Blocksworld (simple) | 35.5%* | 56.6%* | **97.8%*** | 50% | 58% | 58% | 90.2% |
| Blocksworld (hard) | 0% | 0.9%* | 23.65%* | 8% | 48% | 4% | **78%** |
| Floor Tiles | 0% | 6% | 20% | 0% | 4% | – | **36%** |
| Parking | 2% | 8% | 40% | 14% | 28% | – | **52%** |
| Tetris | 0% | 2% | 26% | 0% | 12% | – | **38%** |
| Elevator | 0% | 2% | 36% | 2% | 10% | – | **48%** |
| Barman | 0% | 0% | 10% | 0% | 4% | – | **30%** |

Table 1: Accuracy results across all evaluated domains. Each baseline was evaluated on 50 problem instances per domain, except for GPT-4o + Visual Thinking on Blocksworld (simple), where we evaluated 500 instances (the full PlanBench [56]) and achieved 90.2% (451/500). Baseline results marked with * are taken from [58]. RAP reported 51% accuracy on Blocksworld (simple) using Llama 2 [25].

Unlike prior approaches that augment language models with visual representations for compositional reasoning [66, 30]—which typically provide an initial visual template for the model to iteratively update—Visualizing Thought relies solely on textual descriptions. The model autonomously generates conceptual diagrams from scratch for every state, without any human-provided visual examples or cues, mirroring how humans use imagination to construct mental models from language. Moreover, instead of producing a single static visualization [30, 60], our method generates evolving sequences of intermediate diagrams that illustrate how the LMM's 'model' of the problem evolves with each reasoning step.

Evaluations across multiple challenging PDDL planning domains demonstrate that our method substantially enhances LLMs' combinatorial reasoning capabilities (Tab. 1). On the widely studied Blocksworld domain, from PlanBench [56], our approach delivers performance gains of 43, 64, and 55 percentage points using Claude 3.5 Sonnet [5], Llama 4 Maverick [41], and GPT-4o, respectively (e.g., GPT-4o's accuracy rises from 35.5% to 90.2%). Importantly, we contribute *a new, more difficult planning benchmark* with five additional planning domains—Floor Tiles, Parking, Tetris, Elevator, and Barman—with solution depths designed up to 40. On this benchmark, our method succeeds where base models consistently fail (e.g., 36% vs. 0% in Floor Tiles). Furthermore, Visual Thinking (using GPT-4o) outperforms the reasoning model, o1-preview [45], across all new domains (e.g., 10% vs. 30% in Barman). Finally, compared to strong search-based inference methods such as Graph-of-Thought [7] and RAP [25] (a Monte Carlo Tree Search framework that uses an LLM to build world models and generate plans), our method improves accuracy by at least 22 percentage points while also reducing inference cost by over 30% and latency by more than 25%.

Crucially, our ablation study on the Blocksworld (simple) domain shows that it is the representation of relational information in conceptual diagrams, not merely the encoded content, that drives these gains. Replacing rendered diagrams with their underlying Matplotlib code, which contains the same spatial and relational data, caused accuracy to collapse from 90.2% to 24%, below the GPT-4o single-inference baseline (35.5%) (Table 5). This sharp decline shows that the compact, parallel, multi-dimensional (2D, color-encoded) representation of object interdependencies in diagrams, rather than their sequential form in text or their syntactically cluttered code representation, is what enables more effective reasoning.

To summarize, our contributions include: 1) a cognitively inspired reasoning framework, Visualizing Thought, that enables LMMs to reason with conceptual diagrams autonomously generated from textual descriptions, with no manual engineering or visual templates required for new domains, within a structured graph-based inference process; 2) empirical evidence that representation of information, not just the content, is critical for reasoning, as replacing rendered diagrams with their code containing the same data causes performance to collapse; and 3) extensive evaluations on PlanBench and a new benchmark of five long-horizon planning domains, where our method consistently outperforms strong search-based baselines and reasoning models (on new domains), demonstrating that conceptual diagrams enable solving problems beyond the reach of purely textual (single-inference or search-based) approaches.

## 2 RELATED WORK

**Multimodal Representations for Reasoning**. Several recent works have explored the integration of visual representations with reasoning processes of LLMs and LMMs. Hu et al. [30] equips LMMs with drawing tools to graph equations or mark photorealistic images, but primarily focuses on single-step or shallow problems. Similarly, Wang et al. [60] generates visual aids for spatial reasoning, providing a single refined visualization per problem. In both works, the generated visualizations are typically approximate or augmentations of high-fidelity illustrations rather than conceptual diagrams drawn

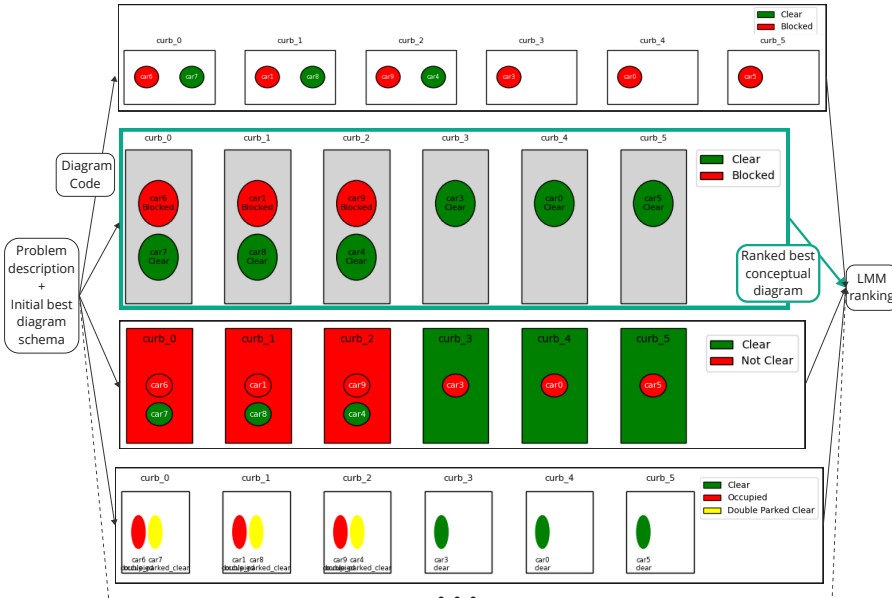

Figure 2: We generate an initial conceptual diagram for each domain by sampling multiple diagram codes. An LMM ranks diagrams based on intuitive and accurate visualization of relational information. The top-ranked diagram's code serves as reference for generating initial and goal state diagrams for all instances. Example shown is the parking domain, where curbs hold up to two cars, and cars can be movable or blocked (double-parked).

using a model-defined mapping of entities to simple shapes and colors. Concurrent work, Wu et al. [66], also generates visual and textual intermediate states but requires conditioning the model on a human-provided initial visual representation that supplements the textual description of the problem. This reliance on externally supplied visualizations could limit generality and applicability to unseen domains.

Our approach differs in several key aspects. First, our method autonomously generates conceptual diagrams directly from textual descriptions, without relying on external visual demonstrations or cues, mirroring human ability to construct mental models from language. Second, rather than producing a single visualization, our approach creates multiple chains of intermediate visual states, enabling parallel multi-hypothesis compositional reasoning through evolving diagrams. Third, our diagrams are conceptual, representing relationships and interactions between entities that are visualized with simple shapes rather than realistic depictions. Finally, our method is applicable to any problem expressible in PDDL format, requiring no domain-specific engineering beyond a textual description. These distinctions collectively enable a more generalizable and flexible form of diagrammatic reasoning, leading to significant performance gains.

In parallel, Table as Thought (TaT) [52] structures intermediate reasoning steps as tabular schemas, showing benefits for planning and math. Similarly, Chain-of-Table (CoTb) [62] maintains an evolving table of intermediate states for table QA and fact-checking. Our diagram schema serves an analogous role: a compact structured intermediate that organizes evolving constraints, but each state representation is extended to a multimodal node combining textual description, diagram schema, and rendered conceptual diagram. As confirmed by our ablations, this structure aids reasoning even when diagrams are not rendered (see "Diagram Schema Only"), while the rendered 2D form provides an additional relational encoding that further boosts accuracy.

Furthermore, efficiency motivations resonate with work like DeepSeek-OCR [63], which compresses long contexts via optical 2D mappin, supporting our finding that diagrammatic intermediates serve as compact, token-efficient carriers of structured state during search. We extend this line by embedding a conceptual diagram within each node and combining beam search with depth-wise backtracking, reducing combinatorial blow-up while retaining recovery from false negatives.

**Search and Verification Inference Strategies**. Recent methods improve reasoning in LLMs by structuring inference into explicit intermediate steps [64, 33] and employing search-based strategies over multiple reasoning paths [61, 68, 25, 16, 7]. For instance, Yao et al. [68] propose Tree-of-Thought (ToT), extending CoT to explore a tree of reasoning paths, and Besta et al. [7] introduce Graph-of-Thought (GoT), representing reasoning as a graph that supports backtracking and aggregation of intermediate steps. Other works apply verification and refinement through iterative feedback [38, 46, 51].

**World Modeling with LLMs**. Recent research explores planning and reasoning using LLMs by implicitly or explicitly constructing world models from text [31, 67, 3, 17, 19, 35, 59]. For instance, RAP [25],

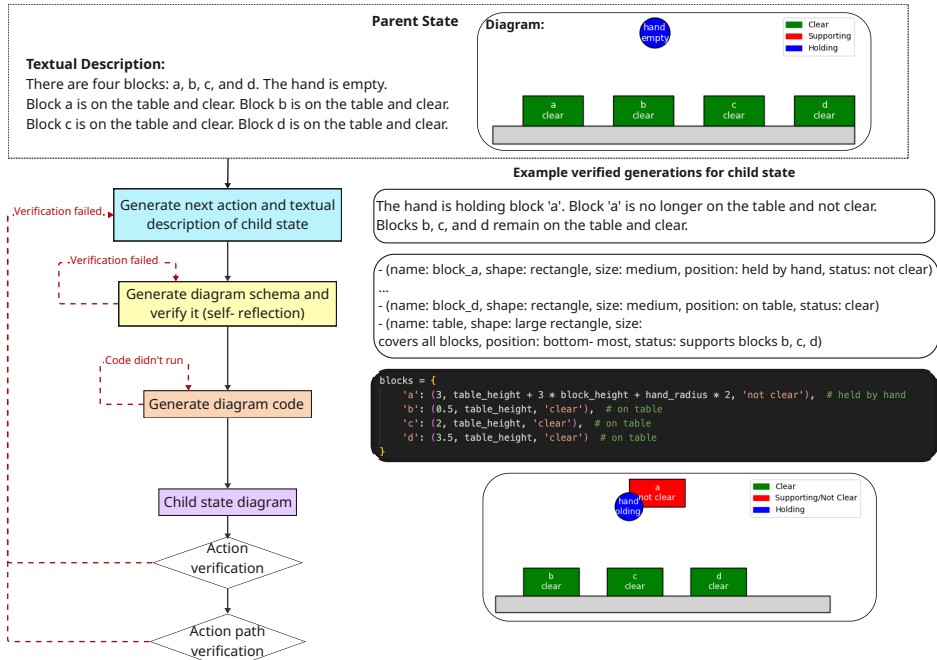

Figure 3: Child state generation pipeline: the LMM selects an action from the parent node, generates a diagram schema and then an executable diagram code, performs self-reflection and verifies that action chosen does not violate any constraints and action path is feasible (example generations are from Blocksworld domain).

which we use as a search-based baseline, employs an LLM as both planner and world modeler, simulating future states via Monte Carlo Tree Search. Our work extends these text-based approaches by enabling LMMs to construct and reason with *diagrammatic world models*—visual schemas of objects and relations encoded through color, shape, and spatial layout. This yields a compact, parallel representation of relational structure that improves performance on complex multi-step planning tasks. In line with Tables as Texts or Images [18], which shows that representational format (text vs. image) critically affects reasoning accuracy, our ablations reveal a sharp accuracy drop (90.2% → 24%) when replacing rendered diagrams with their code—identical in content but not in perceptual form—highlighting that the visual format itself, not just informational content, drives reasoning efficacy.

## 3 METHOD

We propose Visual Thinking, a training-free and model-agnostic framework that integrates textual reasoning with model-generated intermediate conceptual diagrams to enable Large MultiModal Models (LMMs) to solve combinatorial problems, given text-only problem specifications (initial state, goal, admissible actions). Combinatorial problems [65] involve finding a valid sequence of actions from an initial state $s_0$ to a goal state $s_g$, given a finite set of possible actions. Our framework, built upon the Graph-of-Thought (GoT) approach [7], decomposes reasoning into discrete nodes in a structured inference graph. Through this graph, multiple chains of multimodal states are simultaneously explored toward the goal state. Below, we detail our method following the stages illustrated in Fig. 1. The full implementation, with full prompts and outputs, is available in the supplementary material. An analysis of the prompts and an overview of the code structure are provided in Appendix Sec. D and Sec. E, respectively.

**1. Generating Initial Conceptual Diagram for Domain**. Step 1 in Fig. 1. For each new planning domain, the LMM generates a reference conceptual diagram from a random domain instance. This is done entirely in a zero-shot manner, without manual engineering or external visual cues, conditioned only on the problem's textual description. The process begins with the LMM proposing multiple candidate diagram schemas, which it then verifies by iterating through the objects to confirm the accuracy of their shape, color, status, and relative size and position. The LMM ranks these verified schemas on how clearly they represent object relationships, selecting the top one. Using this schema, the model generates several executable Matplotlib diagram codes. Each rendered diagram is then verified to ensure objects are represented accurately and do not overlap. Finally, the LMM ranks these diagrams based on how effectively they visualize the structure and relationships between objects, and the code for the highest-ranked diagram is cached as a reference for generating the diagram code of the initial and goal states of all instances in the domain. See Fig. 2 for example reference diagrams generated for Parking domain.

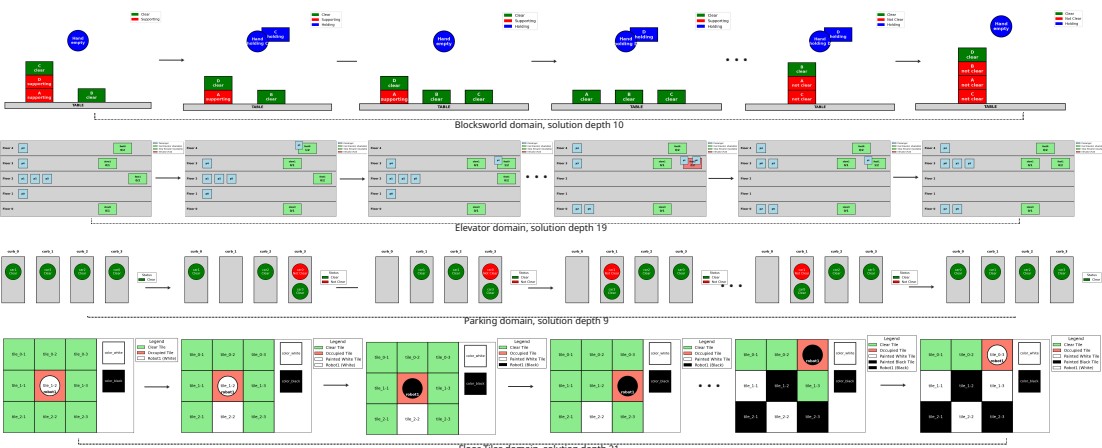

Figure 4: Sequence of intermediate state diagrams in the correct chain, from the initial to the goal state, for one instance across four evaluated domains: Blocksworld, Elevator, Parking, and Floor Tiles (shown top to bottom).

**2. Initial and Goal State Diagram Generation**. Step 2 in Fig. 1. We begin the inference process of each instance by generating diagrams for the initial state $s_0$ and the goal state $s_g$, conditioned on the domain conceptual diagram code obtained in step 1. When generating the diagram code, the LMM is instructed to adhere to how objects and their statuses are visualized in the reference diagram, while accurately initializing the objects according to the specific instance.

**3. Intermediate Child State Generation Pipeline**. Step 3 in Fig. 1. We denote by $s_d$ an intermediate state at depth $d$ of the graph, represented by a combination of: (i) textual description $T(s_d)$; (ii) a diagram $D(s_d)$; and (iii) the action path $A_{0:d}$ from the initial state $s_0$. We iteratively expand the inference graph depth-by-depth in a breadth-first search (BFS) [14] manner, and apply beam search at each depth to select the top-k candidates for further expansion.

From each parent state $s_d$, we sample $n=4$ child states. W.l.o.g., next we describe the generation of a single such child state $s_{d+1}$ (Fig. 3). At each node, the LMM first selects the next candidate action $a_{d+1}$, conditioned on the parent state $s_d$ (which is represented by its diagram, textual description, and action path from initial state). We then generate the textual description $T(s_{d+1})$ of the resulting child state. The candidate action-state pair $(a_{d+1}, T(s_{d+1}))$ is compared to previously generated child states to verify uniqueness. If different, the LMM generates a diagram encoding for $s_{d+1}$, denoted as $E(s_{d+1})$, which is a structured set of textual statements specifying shapes, sizes, positions, statuses (e.g., colors), and textual identifiers for each object in the state (see Fig. 3 for an example). The $E(s_{d+1})$ undergoes a self-reflection verification to ensure consistency with $T(s_{d+1})$ and the action taken, $a_{d+1}$, if failed we regenerate it. Subsequently, Matplotlib code $C(s_{d+1})$ is generated conditioned on $E(s_{d+1})$, $T(s_{d+1})$, and two example diagram codes: the initial state diagram code $C(s_0)$ and the parent state diagram code $C(s_d)$. Code is regenerated if it fails to run.

After generating the child state diagram, we perform two action verifications: (1) a local check that, given the diagram and description of the parent and child state, confirms if the action $a_{d+1}$ complies with domain constraints; and (2) a global check verifying if the entire action path $A_{0:d+1}$ is feasible and efficient for reaching the goal state $s_g$. If either of these checks fails, the child state is marked as invalid.

**4. Beam Search**. Step 4 in Fig. 1. Applying GoT [7] to combinatorial problems by naively expanding all nodes (e.g., BFS) results in exponential growth of the search tree [49]. To mitigate this, we use a method inspired by beam search [37], where first all states at each depth $d$ are expanded, generating up to $N$ child states each. The LMM then ranks all candidate child states at depth $d+1$ based on their proximity to the goal $s_g$, selecting only the top $k=4$ states for further expansion. This depth-wise pruning mitigates the exponential growth problem. Moreover, this ranking system resembles human problem-solving strategies, where shallow state-specific heuristics are employed to estimate how close intermediate states are to the goal [23, 10, 48].

**5. Depth-wise Backtracking**. Step 5 in Fig. 1. Additionally, we implement a depth-wise backtracking mechanism. If all candidate child states at a given depth fail verification, we backtrack to the deepest available ancestor nodes at depth $d_{max}$ and attempt new expansions. We allow up to $B=2$ backtracking attempts to any given depth. If all $B$ attempts at depth $d_{max}$ fail, we mark nodes at that depth as invalid and backtrack further to the next deepest available nodes at depth $d_{max2}$, where $d_{max2} < d_{max}$.

**6. Search Termination**. Step 6 in Fig. 1. The inference process continues iteratively, expanding nodes depth-by-depth, until either the goal state $s_g$ is reached or a predefined computational budget is

| Analysis | Corrects (Accuracy) | Incorrects | Incompletes | Avg Depth | Max Depth | Min Depth | Avg Num States |
|---|---|---|---|---|---|---|---|
| Blocksworld (hard) | 78% | 10% | 12% | 20.15 | 36 | 18 | 177.7 |
| Floor Tiles | 36% | 24% | 40% | 15.88 | 25 | 10 | 244 |
| Parking | 52% | 40% | 8% | 9.6 | 29 | 2 | 149.1 |
| Tetris | 38% | 60% | 2% | 6.45 | 11 | 4 | 46.24 |
| Elevator | 48% | 42% | 10% | 20.89 | 26 | 16 | 160.30 |
| Barman | 30% | 40% | 30% | 24.0 | 27 | 22 | 210.54 |

Table 2: Analysis of domains. 'Corrects': % instances with correct plans; 'Incorrects': % with incorrect plans; 'Incompletes': % terminated due to state budget; 'Avg/Max/Min Depth': avg/max/min number of actions in correct solutions; 'Avg Num States': average number of states generated across all instances of the domain.

exhausted. We set two types of computational limits: (1) a maximum number of generated states (120 states for simpler Blocksworld instances, 450 states for more complex domains), and (2) a maximum depth (28 for simpler Blocksworld instances, 100 for other domains). These limits ensure computational efficiency in terms of inference time and API usage costs. If the goal state is not found within these constraints, the search is marked as incomplete. Goal verification occurs at every state expansion, where the LMM compares the diagram and textual description of the current state against those of the goal state.

In summary, our method leverages structured visual reasoning, self-generated conceptual diagrams, and optimized graph-based inference strategies to efficiently solve combinatorial planning problems with an LMM. Each step of our pipeline is visually illustrated in Figures 1, 2, and 3.

## 4 EVALUATION

To evaluate the proposed approach, we conducted experiments on seven different planning domains, including the popular Blocksworld (simple) [24, 56] and Blocksworld (hard) [58], as well as 5 new domains contributed in this work, prepared using a standard repository of PDDL problem instance generators [1]. Plan correctness was determined with VAL [29]. Experiments were run on a machine with dual-socket Intel Xeon Gold 5220R processors at 2.2 GHz, 35.75 MB L3, 48 cores per node, 8 nodes total.

**Baselines**. We compare our approach against both single-inference and search-based methods. For single-inference baselines, we evaluated GPT-4o, o1-preview [45], and o1-mini [44], each prompted with PDDL instances using templates adapted from [56]. For search-based methods, we evaluate (i) a baseline variant of Graph-of-Thought (GoT), which performs text-only breadth-first search over LLM-generated states; (ii) Optimized GoT, which adds beam search ($k=4$) to GoT to enable exploring deeper solutions within the compute budget; and (iii) RAP [25], which uses Monte Carlo Tree Search with the LLM generating both the world model and the plan. RAP's compute budget is specified by iteration count; we adopted RAP[(20)], the highest budget reported by the authors in their experiments. RAP was originally evaluated with Llama models; we extended it to GPT-4o for direct comparison. However, RAP's implementation relies on hard-coded prompts and domain-specific parsers available only for Blocksworld, limiting applicability to other domains. Together, these baselines test the limits of purely text-based search and single inference and provide strong points of comparison for our diagram-based framework.

**Evaluated Domains**. We evaluated our method on combinatorial planning problems [22] from the International Planning Competition (IPC) [2], expressed in PDDL format [40]. These domains include Blocksworld (simple) and Blocksworld (hard) [24, 58], and five additional IPC domains: Floor Tiles [27], Parking, Tetris [55], Elevator, and Barman. Instances for new domains were generated using standard publicly available IPC generators [1]. For Blocksworld (which involves stacking and unstacking blocks), we used 500 simple instances from PlanBench [56] (3–5 blocks) and 50 harder instances (10–20 blocks, following [58]). Floor Tiles features robots painting tiles on a grid; we generated 50 instances with 2–3 rows, 3–5 columns, and 1–2 robots. The Parking domain involves rearranging cars in curbs, with 50 instances using 4-5 curbs and 4-6 cars. The Tetris domain requires rearranging Tetris tiles on a grid, with 50 instances using $(4 \times 4)$ or $(6 \times 6)$ grids. Lastly, the Elevator domain simulates passenger transport in buildings, with 50 instances using 4-5 floors and 10-12 passengers. Figure 4 shows an example sequence of intermediate state diagrams in the correct plan for a subset of domains. Detailed definitions of each domain are provided in Appendix Sec. C.

**Translating PDDL to Natural Language and Back**. Our method, Visualizing Thought, operates on the natural language description of combinatorial problems. To enable this, we first translate each PDDL domain—the rules and allowed actions—into natural language using a manually engineered five-shot prompt covering five different domains. Each instance, which specifies the initial and goal states, is translated with a one-shot prompt. Our proposed approach then runs entirely on this text representation. After solving the problem, the model's natural language action sequence is translated

| Model | Meeting Planning | Calendar Scheduling |
|---|---|---|
| GPT-3.5 | 19.1% | 19.9% |
| GPT-4 | 47.0% | 41.2% |
| GPT-4o | 45.2% | 43.7% |
| Gemini 1.5 Flash | 23.9% | 34.3% |
| Gemini 1.5 Pro | 39.1% | 48.9% |
| **Visual Thinking (GPT-4o)** | **83.0%** | **91.0%** |

Table 3: **Natural-Plan [69] generalization results.** Exact-match accuracy on 100 instances per domain. Baseline numbers are taken from Zheng et al. [69]. Visual Thinking shows strong transfer to natural-language planning tasks.

| Base LMM | Single-Inference Accuracy | + Visual Thinking |
|---|---|---|
| GPT-4o | 35.5% | 90.2% |
| Claude 3.5 Sonnet | 54.8% | 98.0% |
| Llama 4 (Maverick) | 10.0% | 74.0% |
| Qwen3-VL | 28.0% | 78.0% |

Table 4: **Cross-model generalization results.** Visual Thinking consistently boosts reasoning across proprietary and open-source LMMs, including `Qwen3-VL`.

back into PDDL using a one-shot prompt containing a random (incorrect) plan with correct PDDL syntax. The resulting PDDL plan is then evaluated for correctness using VAL.

## 4.1 Results and Analysis

Our main results are presented in Table 1, comparing Visual Thinking (with GPT-4o) against leading reasoning models (o1-preview, o1-mini) and strong search-based methods (GoT, Optimized GoT, RAP). Visual Thinking substantially improves over base GPT-4o and consistently outperforms all search-based approaches across domains. On Blocksworld (simple), our method achieves 90.2% accuracy, surpassing GoT, Optimized GoT, and RAP, though slightly trailing o1-preview (97.8%). We conjecture this gap is due to the smaller number of entities, which make the world state easier to track and update in text, and the shallow solution depths, which make these instances easier to solve in a single pass. In contrast, on harder domains, including Blocksworld (hard) and the five new domains, our approach shows considerable, generalizable gains. For example, on Blocksworld (hard) we achieve 78%, compared to 23.65% for o1-preview and 4% for RAP. This trend holds across other domains. Standard GoT often fails completely (e.g., 0% on Floor Tiles and Tetris) due to combinatorial explosion exhausting the budget, and while Optimized GoT mitigates this with beam search, its performance still lags well behind our visual approach. These findings highlight how diagram-based reasoning enables models to capture and analyze complex relational structures more efficiently than purely textual inference.

Table 2 provides further insight into our method's performance on the more challenging domains. Despite significantly deeper solution paths (instances were designed with solution paths of up to 40), our method successfully generates correct plans with as many as 36 sequential actions (Blocksworld (hard)). The primary limitation of our method observed in these experiments is the number of incomplete searches (e.g., 40% incomplete in Floor Tiles, 30% in Barman), which arise either when invalid actions are later rejected by local verification using state diagrams, or when inefficient actions that fail to advance toward the goal are pruned by the global check, both leading to exhaustion of the computational budget. We also observe the highest incorrect rate (60%) on the Tetris domain, primarily due to inherently high branching factor in this domain (up to 24 possible actions per state) and complex action parameterization—each action can require up to 7 parameters detailing positions of all sub-tiles, compared to simpler domains like Blocksworld, where actions typically require only 1-2 parameters. Conversely, our largest margin over o1-preview occurs in the Barman domain (30% vs. 10%), likely because diagrammatic representations capture the high number of object statuses and interactions per state in this domain more effectively than text alone.

**Model Generalization**. To assess the generalizability of our framework, we evaluated it on other state-of-the-art LMMs using 50 instances from the Blocksworld (simple) domain. With Llama 4, our method increased accuracy from a baseline of 10% (single inference) to 74%, an over 7x improvement in accuracy. The improvement was also observed using Claude 3.5 Sonnet, where accuracy increased from 54.8% (using zero shot single inference as reported in [58]) to 98%, achieving state-of-the-art performance on this benchmark. These substantial gains demonstrate that the benefits of our approach are not tied to a specific model architecture but stem from the fundamental advantage of using model-generated conceptual diagrams as a reasoning medium.

| Ablation | Corrects (Accuracy) | Incorrects | Incompletes | Avg Depth | Max Depth | Avg Num States |
|---|---|---|---|---|---|---|
| Visual Thinking + GPT-4o | 90.2% | 6.4% | 3.4% | 10.07 | 28 | 38.89 |
| No Diagram (Optimized GoT) | 58% | 36% | 6% | 8.28 | 18 | 25.86 |
| No Diagram Schema | 72% | 20% | 8% | 7.41 | 18 | 46.2 |
| Diagram Schema Only (no rendered diagram) | 66% | 28% | 6% | 8.04 | 16 | 48.6 |
| No Code Execution | 24% | 66% | 10% | 7.82 | 22 | 32.7 |
| 1-Branching Factor | 52% | 4% | 44% | 6.38 | 16 | 23.9 |
| 2-Branching Factor | 70% | 6% | 24% | 8.29 | 24 | 26.27 |
| No Backtracking | 62% | 4% | 34% | 6.06 | 14 | 22.12 |
| No Beam Search | 72% | 4% | 24% | 6.61 | 12 | 58.31 |

Table 5: Ablation Study Results. 'Corrects' is our main accuracy metric. See Tab. 2 for columns notation.

**Runtime and Cost Analysis**. We analyzed the runtime and API costs of our method and all other search-based baselines on 20 instances per domain. On the Blocksworld (simple) domain, our method had a median runtime of 381 (∼6 minutes) seconds and a median cost of $1.04 per instance. For more complex domains, the median runtime was 1038 seconds (∼17 minutes) with median cost of $2.98 per instance. Our approach is significantly more efficient than other search-based methods. On average, it was 31% faster and 36% cheaper than the GoT baseline, and 46% faster and 52% cheaper than RAP[20]+GPT-4o across all domains, while achieving substantially higher accuracy. Compared to the text-only Optimized GoT, incorporating diagrams added 213 seconds in latency and $0.71 on average, measured across all domains, but this overhead yielded a 30 percentage point accuracy gain.

## 4.2 ABLATION STUDIES

To systematically evaluate the contributions of different components of our framework, we conducted ablation studies on 50 instances from the Blocksworld (simple) domain. We examined the impact of state diagrams, diagram schema, diagram code execution, different branching factors, and inference optimizations (beam search and backtracking). Table 5 summarizes the results of these experiments.

**Impact of Various Components of State Diagram Generation**. We first evaluated the role of state diagrams by removing them entirely from the inference pipeline ("No Diagram"), yielding a text-only optimized Graph-of-Thought approach. This caused accuracy to drop from 90.2% to 58%, underscoring the critical role diagrams play in succinctly representing relational information. Moreover, the average solution depth of correctly solved instances decreased from 10.07 to 8.28, indicating that without diagrams, the model struggled on more complex problems requiring deeper reasoning. In a second experiment, we removed the diagram schema ("No Diagram Schema") from the child-state generation pipeline, instead inferring diagram code directly from textual descriptions of states. Accuracy dropped from 90.2% to 72%, a smaller decline than removing diagrams entirely—showing that diagram schemas further help extract relational information from text, enabling more accurate diagram generation.

Finally, we tested removing the rendered diagrams, providing only the Matplotlib code ("No Code Execution") when generating the next action. This resulted in the most significant performance drop (90.2% to 24%), even below the GPT-4o baseline (35.5%), clearly demonstrating that even though the code encodes the same spatial and relational information as the diagram, the way this information and the interdependencies between objects are represented is crucial for model performance. Using the diagram code directly distracts the model and impairs reasoning, aligning with prior findings that extraneous details negatively impact model performance [50, 36]. These results reinforce the importance of diagrams as compact, intuitive representations that facilitate rapid perceptual inference and clear relational reasoning [34, 54, 26], and that it is the representation of information in a multi-dimensional (2D, color-encoded) format that significantly aids understanding the interdependencies and reasoning in models.

**Impact of Branching Factor**. We next investigated the effect of branching factor of the inference graph (the number of candidate child states generated per state) on performance. Reducing the branching factor from 4 (our default) to 2 ("2-Branching Factor") decreased performance from 90.2% to 70%, primarily due to a sharp increase in incomplete searches (24% vs. 3.4%). This suggests that exploring the third or fourth candidate child states, generated at higher temperatures, is frequently necessary to find the correct solution path. Thus, reducing branching factor limits diversity in candidate state generations, leading to more incomplete searches. Additionally, the average depth of correctly solved instances decreased from 10.07 to 8.29, indicating difficulty solving problems with deeper solution depths. Further reducing the branching factor to 1 ("1-Branching Factor"), effectively converting our graph-based inference

into a multimodal Chain-of-Thought approach (with diagrams), caused performance to drop even further to 52%. Despite this decline, performance remained well above the GPT-4 baseline (35.5%), underscoring the value of diagrams in improving LMM reasoning even without extensive search.

**Impact of Inference Optimizations**. Finally, we evaluated the importance of our inference optimizations (backtracking and beam search) on top of the multimodal inference graph. Removing backtracking ("No Backtracking")—yielding a tree-of-thought method with beam search—reduced accuracy from 90.2% to 62%, primarily due to a sharp rise in incomplete searches (34% vs. 3.4%). This occurs because LMM verification steps occasionally produce false negatives, incorrectly invalidating correct states. Without backtracking, the model cannot recover from these errors, leading to incomplete searches as no validated nodes remain at the frontier of the search graph for further expansion.

Similarly, removing beam search ("No Beam Search") lowered accuracy to 72%, with the incomplete search rate increasing to 24%. In this case, incompletes stem from exponential growth in the search space, causing the model to exhaust its computational budget (i.e., the maximum number of states generated) before reaching the goal. Indeed, the average number of generated states increased significantly (58.31 vs. 38.89), underscoring the critical role of beam search in managing combinatorial explosion. Both optimizations are essential for solving deeper combinatorial problems, as shown by the reduced average correct solution depth without them (6.61 without beam search, 6.06 without backtracking, vs. 10.07 with both). These results demonstrate that backtracking and beam search are complementary and crucial for efficient graph-based combinatorial planning.

## 5 CONCLUSION

**Contributions**. In this paper, we introduced Visual Thinking, a framework that enables LMMs to solve combinatorial problems by reasoning with conceptual diagrams alongside text. Our contributions are: (i) a cognitively inspired method that autonomously generates conceptual diagrams directly from natural language problem descriptions, requiring no human input for new domains; (ii) a multimodal Graph-of-Thought framework that structures reasoning as sequences of intermediate textual and visual states, integrating beam search and backtracking for efficient long-horizon search; (iii) extensive empirical evidence showing substantial performance gains over single-inference LMMs, specialized reasoning models, and strong search-based baselines across various planning benchmarks; and (iv) ablation results demonstrating that representation of information is critical—reasoning improves when relational information and interdependencies are encoded in diagrams, not merely present in text or code format.

**Limitations**. As with any search-based inference method, our framework incurs additional computational cost and inference time to explore multiple reasoning trajectories and also to generate visual representations. However, these overheads are manageable in practice; importantly, our approach remains more efficient than prior text-only search methods such as Graph-of-Thought and RAP. Another limitation concerns scope: Visual Thinking was applied to combinatorial planning problems—a core area of computer science with significant real-world applications, such as warehouse optimization, logistics, and scheduling—chosen because their state-based structure makes diagram progression more straightforward. Future work involves extending conceptual diagram generation to more open-ended problems.

**Future Work**. This work demonstrates a path for LMMs to move beyond purely textual reasoning toward a more powerful, human-like process that integrates visual abstractions. Future work can extend this framework beyond combinatorial planning to more abstract domains, enabling LMMs to produce multimodal outputs such as software architecture diagrams, figures visualizing scientific hypotheses and causal dependencies, or tailored visual aids for educational contexts. Such conceptual diagrams can enhance both model performance and human–AI interaction (e.g., easier verification of code behavior through generated architecture diagrams). This ability to reason and communicate about complex structures multimodally is essential for building AI capable of tackling scientific, creative, and planning tasks.

## 6 REPRODUCIBILITY STATEMENT

An anonymous supplementary repository includes the full inference code, prompt templates, and scripts used in our experiments, along with the PDDL instances of the new planning benchmark created for the five new domains and sample run results. The end-to-end pipeline is specified in Section 3 and summarized in Figure 1. Dataset sources and instance-generation parameters are detailed in Section 4; formal task and action definitions required to regenerate the PDDL domains are provided in Appendix C; and prompt templates are listed in Appendix D. Experimental setup, baselines, and evaluation protocols are described in Sections 4 and 4.1, with ablations in Section 4.2 and comprehensive results in Tables 1 and 5. Code organization and configuration files are documented in Appendix E.

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

APPENDIX

## A    EXAMPLE DIAGRAM GENERATIONS FROM NATURAL-PLAN

We present two illustrative tasks from the Natural-Plan benchmark [69] and the sequences of conceptual diagrams produced by Visual Thinking.

### A.1    CALENDAR SCHEDULING

**Problem.** You are an expert at scheduling meetings. You are given a few constraints on the existing schedule of each participant, the meeting duration, and possibly some preferences on the meeting time. Note there exists a solution that works with existing schedule of every participant. Here are a few example tasks and solutions:

TASK: You need to schedule a meeting for Michelle, Steven and Jerry for one hour between the work hours of 9:00 to 17:00 on Monday.

Here are the existing schedules for everyone during the day: Michelle has meetings on Monday during 11:00 to 12:00; Steven has blocked their calendar on Monday during 9:00 to 9:30, 11:30 to 12:00, 13:30 to 14:00, 15:30 to 16:00; Jerry has blocked their calendar on Monday during 9:00 to 9:30, 10:00 to 11:00, 11:30 to 12:30, 13:00 to 14:30, 15:30 to 16:00, 16:30 to 17:00;

Find a time that works for everyone's schedule and constraints.

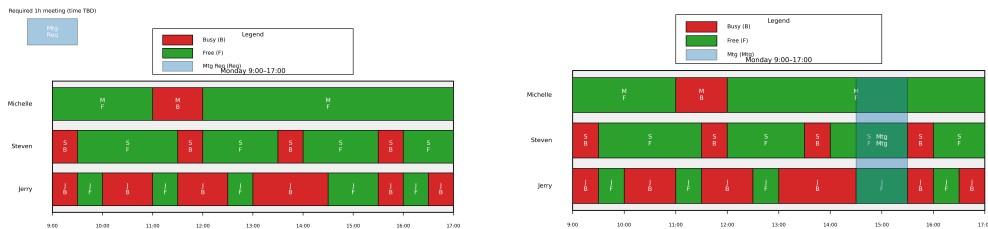

Figure 5: Chain of conceptual diagrams produced for an example calender-scheduling instance.

### A.2    MEETING PLANNING

**Problem.** You are visiting San Francisco for the day and want to meet as many friends as possible. Solve the problem by considering various different schedules and picking the best one to optimize your goals.

Travel distances (in minutes): Fisherman's Wharf to Nob Hill: 11. Nob Hill to Fisherman's Wharf: 11.

Constraints: You arrive at Fisherman's Wharf at 9:00AM. Kenneth will be at Nob Hill from 2:15PM to 7:45PM. You'd like to meet Kenneth for a minimum of 90 minutes.

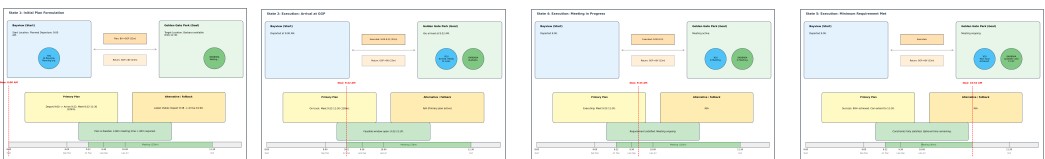

Figure 6: Chain of conceptual diagrams produced for an example meeting-planning instance.

## B    EXAMPLE DIAGRAM SCHEMAS

Below, we provide two example full diagram schemas generated for the initial state of one instance from the Parking domain and another from the Floor Tiles domain.

## B.1 PARKING DOMAIN

**Problem.** The "parking" domain involves cars and curbs. Each car can be parked at a curb, or it can line up behind another car (i.e. double park behind a car X which is parked at curb N). • Clear Car: Means that there is no car behind this particular car, and the car itself is free to move. The car can not move if another car is double parked behind it. • Clear curb: Means that the curb is empty (no car is currently occupying it), so a car could move into that curb.

Actions:

1) move-curb-to-curb

Example: (move-curb-to-curb carX curbFrom curbDest) • Purpose: "carX" which was the first car on "curbFrom" moves to an empty "curbDest" costing 1 unit.

• Preconditions: 1) The carX itself has no car behind it (carX is clear). 2) The destination curb, curbDest, is empty (clear). 3) carX is specifically parked on curbFrom. • Effects: 1) carX vacates curbFrom, making curbFrom empty (clear). 2) carX occupies curbDest, which is no longer clear. 3) carX remains clear.

2) move-curb-to-car

Example: (move-curb-to-car carX curbFrom carAhead) • Purpose: "carX" which was the only car on "curbFrom" moves to double park behind "carAhead" costing 1 unit.

• Preconditions: 1) The carX itself is clear, and has no other car behind it. 2) The destination car, carAhead, is clear (i.e, carAhead is the only car in its curb and has no car behind it). 3) carX is currently parked on curbFrom. • Effects: 1) carX leaves curbFrom, making curbFrom empty (clear). 2) carX is now behind carAhead, so carAhead is no longer clear (carX is below carAhead, carX is green and clear, carAhead is red and not clear and above carX). 3) carX remains clear.

3) move-car-to-curb

Example: (move-car-to-curb carX carAhead curbDest) • Purpose: "carX" which was behind "carAhead" moves from behind carAhead to occupy an empty curb, curbDest, costing 1 unit.

• Preconditions: 1) carX has no cars behind it (clear). 2) curbDest is clear (no car is occupying curbDest). 3) carX is currently behind carAhead (i.e, carX is below carAhead, carX is green and clear, carAhead is red and not clear and above carX). • Effects: 1) carX leaves from behind carAhead, making carAhead clear and green. 2) carX occupies curbDest, which is no longer empty. 3) carX remains clear.

4) move-car-to-car

Example: (move-car-to-car carX carFrom carDest) • Purpose: "carX," which is currently behind "carFrom" moves to double park behind "carDest" costing 1 unit.

• Preconditions: 1) carX has no cars behind it (it is clear) (carX is below carFrom, carX is green and clear, carFrom is red and not clear and above carX). 2) carDest is clear and green (no car is behind it) and is the only car parked at a curb. 3) carX is currently behind carFrom. • Effects: 1) carX is no longer behind carFrom ; carFrom becomes clear and green and the only car at the curb. 2) carX is now behind carDest, so carDest is no longer clear (carX is below carDest, carX is green and clear, carDest is red and not clear and above carX). 3) carX remains clear.

In the initial state, the parking configuration is as follows:- At curb 0, car 6 is parked, and car 7 is double parked behind it. Car 7 is clear.- At curb 1, car 1 is parked, and car 8 is double parked behind it. Car 8 is clear.- At curb 2, car 9 is parked, and car 4 is double parked behind it. Car 4 is clear.- At curb 3, car 3 is parked and is clear.- At curb 4, car 0 is parked and is clear.- At curb 5, car 5 is parked and is clear.- At curb 6, car 2 is parked and is clear.

The goal state involves arranging the cars as follows: Car 6 is parked at curb 6, car 5 is parked at curb 5, car 4 is parked at curb 4, car 3 is parked at curb 3, car 2 is parked at curb 2 with car 9 double parked behind it, car 1 is parked at curb 1 with car 8 double parked behind it, and car 0 is parked at curb 0 with car 7 double parked behind it.

**Diagram schema:**

(text/identifier: curb_0, shape: rectangle, size: medium, position: leftmost in the curb grid, status: contains car2 with car1 double parked behind it)

(text/identifier: curb_1, shape: rectangle, size: medium, position: to the right of curb_0, status: contains car0, clear)

(text/identifier: curb_2, shape: rectangle, size: medium, position: to the right of curb_1, status: contains car3, clear)

(text/identifier: curb_3, shape: rectangle, size: medium, position: to the right of curb_2, status: empty, clear)

(text/identifier: car2, shape: circle, size: small, position: inside curb_0, status: parked, not clear)

(text/identifier: car1, shape: circle, size: small, position: behind car2 in curb_0, status: double parked, clear)

(text/identifier: car0, shape: circle, size: small, position: inside curb_1, status: parked, clear)

(text/identifier: car3, shape: circle, size: small, position: inside curb_2, status: parked, clear)

## B.2 FLOOR TILES DOMAIN

**Problem.** The domain "floor-tile" is about painting a grid of tiles with a set of robots. The robot can hold exactly one color at a time and can move around the grid of tiles one tile at a time. A robot can paint a tile with the color it is currently holding, provided that the tile is "clear" (i.e., not painted and not occupied by the robot) and the tile is accessible to the robot (i.e., the tile to be painted is above or below the current position of the robot). There are 7 actions allowed:

1) change-color

Example usage: (robot1 c c2) Meaning: Robot robot1 changes its paint gun color from c to c2.

• Purpose: Allows the robot to switch the color it is currently holding to another color that is available. • Precondition: – The robot must already have a color c. – The new color c2 must be available. • Effect: – The robot no longer has color C, but now has color C2.

2) paint-up

Example usage: (paint-up robot1 tile_(x-1)-y tile_x-y c) Meaning: Robot robot1, standing on tile tile_x-y, paints the tile above it, tile_(x-1)-y, with the color c, which it is holding. • Purpose: Paints the tile directly above the robot's current position with the robot's current color.

• Precondition: - The robot must be standing on tile_x-y. - Tile tile_(x-1)-y exists and it is directly above tile_x-y where the robot is standing currently (i.e., this tile must be in the same column as the robot and in the row above the robot). - Tile tile_(x-1)-y is clear (i.e., it has not been painted and is not occupied by the robot). – The robot must be holding some color C. • Effect: – Tile tile_(x-1)-y is painted with color c. - Tile tile_(x-1)-y is no longer "clear."

3) paint-down

Example usage: (paint-down robot1 tile_(x+1)-y tile_x-y c) Meaning: Robot robot1, standing on tile tile_x-y, paints the tile below it, tile_(x+1)-y, with the color c, which it is holding. • Purpose: Paints the tile directly below the robot's current position with the robot's current color.

• Precondition: – The robot must be standing on tile_x-y. – Tile tile_(x+1)-y exists and it is directly below tile_x-y (i.e., this tile must be in the same column as the robot and in the row below the robot). – Tile tile_(x+1)-y is clear (i.e., it has not been painted and is not occupied by the robot). – The robot must be holding some color c. • Effect: – Tile tile_(x+1)-y is painted with color c. – Tile tile_(x+1)-y is no longer "clear."

4) up

Example usage: (up robot1 tile_x-y tile_(x-1)-y) Meaning: Robot robot1, standing on tile tile_x-y, moves to the tile directly above it, tile_(x-1)-y. • Purpose: Moves the robot from its current position to the tile directly above it.

• Precondition: - The robot must be standing on tile_x-y. - Tile tile_(x-1)-y exists and is directly above tile_x-y (i.e., in the same column and in the row above). - Tile tile_(x-1)-y must be clear (i.e., it is not painted or occupied). • Effect: - The robot is now standing on tile_(x-1)-y. - Tile tile_x-y becomes clear. - Tile tile_(x-1)-y is no longer clear.

5) down

Example usage: (down robot1 tile_x-y tile_(x+1)-y) Meaning: Robot robot1, standing on tile tile_x-y, moves to the tile directly below it, tile_(x+1)-y. • Purpose: Moves the robot from its current position to the tile directly below it.

• Precondition: - The robot must be standing on tile_x-y. - Tile tile_(x+1)-y exists and is directly below tile_x-y (i.e., in the same column and in the row below). - Tile tile_(x+1)-y must be clear (i.e., it is not painted or occupied). • Effect: - The robot is now standing on tile_(x+1)-y. - Tile tile_x-y becomes clear. - Tile tile_(x+1)-y is no longer clear.

6) right

Example usage: (right robot1 tile_x-y tile_x-(y+1)) Meaning: Robot robot1, standing on tile tile_x-y, moves to the tile directly to its right, tile_x-(y+1). Purpose: Moves the robot from its current position to the tile directly to its right.

• Precondition: - The robot must be standing on tile_x-y. - Tile tile_x-(y+1) exists and is directly to the right of tile_x-y (i.e., in the same row and in the column to the right). - Tile tile_x-(y+1) must be clear (i.e., it is not painted or occupied). • Effect: - The robot is now standing on tile_x-(y+1). - Tile tile_x-y becomes clear. - Tile tile_x-(y+1) is no longer clear.

7) left

Example usage: (left robot1 tile_x-y tile_x-(y-1)) Meaning: Robot robot1, standing on tile tile_x-y, moves to the tile directly to its left, tile_x-(y-1). • Purpose: Moves the robot from its current position to the tile directly to its left.

• Precondition: - The robot must be standing on tile_x-y. - Tile tile_x-(y-1) exists and is directly to the left of tile_x-y (i.e., in the same row and in the column to the left). - Tile tile_x-(y-1) must be clear (i.e., it is not painted or occupied). • Effect: - The robot is now standing on tile_x-(y-1). - Tile tile_x-y becomes clear. - Tile tile_x-(y-1) is no longer clear.

The problem instance describes a scenario where there are 20 distinct floor tiles arranged in a 5×4 layout (we have rows 0, 1, 2, 3, 4 and columns 1, 2, 3, 4), labeled tile_0-1 (located in the top left corner of the grid) through tile_4-4 (located in the bottom right corner of the grid). Two robots (robot1 and robot2) are present, each able to hold exactly one color of paint at a time. The two paint colors available are white and black.

Robot Positions and Paint: - Robot1 begins on the tile labeled tile_1-1 and is currently holding the color white. - Robot2 begins on the tile labeled tile_1-2 and is currently holding the color black.

Available Colors: Both white and black are available to be switched to if needed.

Tiles with Clear Status: - Every tile in the grid is clear except for tile_1-1 (occupied by robot1) and tile_1-2 (occupied by robot2). - Thus, tile_0-1, tile_0-2, tile_0-3, tile_0-4 in row 0 - tile_1-3, tile_1-4 in row 1 - tile_2-1, tile_2-2, tile_2-3, tile_2-4 in row 2 - tile_3-1, tile_3-2, tile_3-3, tile_3-4 in row 3 - and tile_4-1, tile_4-2, tile_4-3, tile_4-4 in row 4 are all not painted and not occupied.

• The aim is to have every listed tile in rows 1 through 4 (i.e., tile_1-1, tile_1-2, tile_1-3, tile_1-4, and so forth down to tile_4-4) painted with an alternating pattern of white and black. Specifically:

– Row 1 (tiles tile_1-1 through tile_1-4): white, black, white, black.

– Row 2 (tiles tile_2-1 through tile_2-4): black, white, black, white.

– Row 3 (tiles tile_3-1 through tile_3-4): white, black, white, black.

– Row 4 (tiles tile_4-1 through tile_4-4): black, white, black, white.

**Diagram Schema:**

(text/identifier: tile_0-1,shape: rectangle,size: small,position: top-left corner of the grid,status: occupied by robot1 holding color white)(text/identifier: tile_0-2,shape: rectangle,size: small,position: to the right of tile_0-1,status: clear)

(text/identifier: tile_0-3,shape: rectangle,size: small,position: to the right of tile_0-2,status: clear)

(text/identifier: tile_1-1,shape: rectangle,size: small,position: directly below tile_0-1,status: clear)

(text/identifier: tile_1-2,shape: rectangle,size: small,position: to the right of tile_1-1,status: clear)

(text/identifier: tile_1-3,shape: rectangle,size: small,position: to the right of tile_1-2,status: clear)

(text/identifier: tile_2-1,shape: rectangle,size: small,position: directly below tile_1-1,status: clear)

(text/identifier: tile_2-2,shape: rectangle,size: small,position: to the right of tile_2-1,status: clear)

(text/identifier: tile_2-3,shape: rectangle,size: small,position: to the right of tile_2-2,status: clear)

(text/identifier: robot1,shape: circle,size: smaller than a tile,position: on tile_0-1,status: holding color white)

(text/identifier: color_white,shape: rectangle,size: smaller than each tile,position: available for switching,status: available)

(text/identifier: color_black,shape: rectangle,size: smaller than each tile,position: available for switching,status: available)

## C   DOMAIN DEFINITIONS

This section provides formal definitions for the five planning domains introduced in our benchmark. For each domain, we describe the main objective and provide a detailed specification of the available actions, including their purpose, preconditions, and effects.

### C.1   BARMAN DOMAIN

The Barman domain models the task of a bartender preparing cocktails. The agent must use two hands to manipulate containers (shots, shakers), ingredients from dispensers, and mix them to create specific cocktails. The state of each object includes its location (on table or held), contents, cleanliness, and for shakers, fill level and whether it has been shaken.

ACTIONS

1. **grasp(hand, container)**: An empty hand picks up a container from the table.
   - **Preconditions**: The container is on the table; the hand is empty.
   - **Effects**: The container is no longer on the table; the hand now holds the container and is no longer empty.
2. **leave(hand, container)**: A hand places a held container onto the table.
   - **Preconditions**: The hand is holding the container.
   - **Effects**: The container is now on the table; the hand becomes empty.
3. **fill-shot(shot, ingredient, hand1, hand2, dispenser)**: Fills a clean, empty shot with an ingredient.
   - **Preconditions**: hand1 holds the shot; hand2 is empty; the dispenser provides the ingredient; the shot is empty and clean.
   - **Effects**: The shot now contains the ingredient and is no longer empty or clean (it becomes used).
4. **refill-shot(shot, ingredient, hand1, hand2, dispenser)**: Refills a used shot with the same ingredient it previously held.
   - **Preconditions**: hand1 holds the shot; hand2 is empty; the dispenser provides the ingredient; the shot is empty and was previously used with this ingredient.
   - **Effects**: The shot now contains the ingredient and is no longer empty.
5. **empty-shot(hand, shot, beverage)**: Empties the contents of a shot.
   - **Preconditions**: The hand is holding the shot; the shot contains the beverage.
   - **Effects**: The shot becomes empty.
6. **clean-shot(shot, beverage, hand1, hand2)**: Cleans a used, empty shot.
   - **Preconditions**: hand1 holds the shot; hand2 is empty; the shot is empty and was previously used with the beverage.
   - **Effects**: The shot becomes clean and is no longer considered used.
7. **pour-shot-to-clean-shaker(shot, ingredient, shaker, hand, level-prev, level-next)**: Pours an ingredient from a shot into a clean, empty shaker.
   - **Preconditions**: The hand holds the shot containing the ingredient; the shaker is empty and clean; the shaker is at level-prev.
   - **Effects**: The shot becomes empty; the shaker now contains the ingredient, is no longer empty or clean, becomes unshaken, and its fill level increases to level-next.

8. **pour-shot-to-used-shaker(shot, ingredient, shaker, hand, level-prev, level-next)**: Adds a second ingredient to an unshaken shaker.
   - **Preconditions**: The hand holds the shot containing the ingredient; the shaker is unshaken and contains one ingredient; the shaker is at level-prev.
   - **Effects**: The shot becomes empty; the shaker now contains the additional ingredient; the shaker's fill level increases to level-next.

9. **empty-shaker(hand, shaker, cocktail, level-prev, level-next)**: Empties a shaken cocktail from the shaker.
   - **Preconditions**: The hand holds the shaker; the shaker contains a shaken cocktail; the shaker is at level-prev.
   - **Effects**: The shaker becomes empty and unshaken; its fill level resets to level-next (empty).

10. **clean-shaker(hand1, hand2, shaker)**: Cleans an empty shaker.
    - **Preconditions**: hand1 holds the shaker; hand2 is empty; the shaker is empty.
    - **Effects**: The shaker becomes clean.

11. **shake(cocktail, ing1, ing2, shaker, hand1, hand2)**: Mixes two ingredients in a shaker to create a cocktail.
    - **Preconditions**: hand1 holds the shaker; hand2 is empty; the shaker contains exactly ing1 and ing2; the shaker is unshaken.
    - **Effects**: The shaker becomes shaken; it now contains the resulting cocktail instead of the separate ingredients.

12. **pour-shaker-to-shot(cocktail, shot, hand, shaker, level-prev, level-next)**: Serves a shaken cocktail from a shaker into a shot.
    - **Preconditions**: The hand holds the shaker containing the shaken cocktail; the shot is empty and clean; the shaker is at level-prev.
    - **Effects**: The shot now contains the cocktail and is no longer empty or clean; the shaker's fill level decreases to level-next.

## C.2 ELEVATOR DOMAIN

The Elevator domain involves operating a set of elevators (fast and slow) to transport passengers between floors in a building. Each elevator has a specific capacity and can only access a defined set of floors. The goal is to move all passengers from their origin floors to their destination floors.

ACTIONS

1. **move-up-slow(elevator, floor-from, floor-to)**: Moves a slow elevator up.
   - **Preconditions**: The elevator is at floor-from; floor-to is above floor-from; the elevator can reach floor-to.
   - **Effects**: The elevator is now at floor-to.

2. **move-down-slow(elevator, floor-from, floor-to)**: Moves a slow elevator down.
   - **Preconditions**: The elevator is at floor-from; floor-to is below floor-from; the elevator can reach floor-to.
   - **Effects**: The elevator is now at floor-to.

3. **move-up-fast(elevator, floor-from, floor-to)**: Moves a fast elevator up.
   - **Preconditions**: The elevator is at floor-from; floor-to is above floor-from; the elevator can reach floor-to.
   - **Effects**: The elevator is now at floor-to.

4. **move-down-fast(elevator, floor-from, floor-to)**: Moves a fast elevator down.
   - **Preconditions**: The elevator is at floor-from; floor-to is below floor-from; the elevator can reach floor-to.
   - **Effects**: The elevator is now at floor-to.

5. **board(passenger, elevator, floor, count-prev, count-next)**: A passenger boards an elevator.
   - **Preconditions**: The passenger and elevator are at the same floor; the elevator's passenger count is count-prev; the elevator has capacity for another passenger.
   - **Effects**: The passenger is now on board the elevator; the elevator's passenger count becomes count-next.

6. **leave(passenger, elevator, floor, count-prev, count-next)**: A passenger leaves an elevator.

- **Preconditions**: The `passenger` is on board the `elevator`; the `elevator` is at the specified `floor`; the elevator's passenger count is `count-prev`.
- **Effects**: The `passenger` is now at the `floor`; the elevator's passenger count becomes `count-next`.

### C.3 PARKING DOMAIN

The Parking domain involves rearranging cars parked at curbs. Each curb can hold at most two cars: one parked at the curb and one double-parked behind it. A car cannot move if another car is parked behind it.

KEY PREDICATES

- **clear(car)**: True if no car is double-parked behind this car.
- **clear(curb)**: True if the curb is empty.

ACTIONS

1. **move-curb-to-curb(car, curb-from, curb-to)**: A single-parked car moves to an empty curb.
   - **Preconditions**: car is at `curb-from`; car is clear; `curb-to` is clear.
   - **Effects**: `curb-from` becomes clear; `car` is now at `curb-to`, which is no longer clear.
2. **move-curb-to-car(car-move, curb-from, car-ahead)**: A single-parked car double-parks behind another car.
   - **Preconditions**: `car-move` is at `curb-from`; `car-move` is clear; `car-ahead` is clear.
   - **Effects**: `curb-from` becomes clear; `car-move` is now behind `car-ahead`; `car-ahead` is no longer clear.
3. **move-car-to-curb(car-move, car-ahead, curb-to)**: A double-parked car moves to an empty curb.
   - **Preconditions**: `car-move` is behind `car-ahead`; `car-move` is clear; `curb-to` is clear.
   - **Effects**: `car-ahead` becomes clear; `car-move` is now at `curb-to`, which is no longer clear.
4. **move-car-to-car(car-move, car-from, car-to)**: A double-parked car moves to double-park behind a different car.
   - **Preconditions**: `car-move` is behind `car-from`; `car-move` is clear; `car-to` is clear.
   - **Effects**: `car-from` becomes clear; `car-move` is now behind `car-to`; `car-to` is no longer clear.

### C.4 TETRIS DOMAIN

The Tetris domain involves moving and rotating Tetris pieces on a grid. Pieces can be one-square, two-square straight, or three-square L-shaped. A piece can only move or rotate into adjacent positions that are clear (empty).

ACTIONS

1. **move_square(pos-from, pos-to, piece)**: Moves a one-square piece.
   - **Preconditions**: `piece` occupies `pos-from`; `pos-to` is clear; `pos-from` and `pos-to` are adjacent.
   - **Effects**: `pos-from` becomes clear; `pos-to` is now occupied by `piece`.
2. **move_two(pos-old, pos-pivot, pos-new, piece)**: Moves or rotates a two-square piece.
   - **Preconditions**: `piece` occupies `pos-old` and `pos-pivot`; `pos-new` is clear; `pos-pivot` and `pos-new` are adjacent.
   - **Effects**: `pos-old` becomes clear; `pos-new` is now occupied by `piece`.
3. **move_l_right(pA, pB, pC, pD, pE, pMid, piece)**: Moves or rotates an L-piece right.
   - **Preconditions**: `piece` occupies `pA, pB, pC`; `pD, pE` are clear; positions are correctly adjacent for a rightward move.
   - **Effects**: `pA, pB` become clear; `pD, pE` are now occupied by `piece`.
4. **move_l_left(pA, pB, pC, pD, pE, piece)**: Moves or rotates an L-piece left.
   - **Preconditions**: `piece` occupies `pA, pB, pC`; `pD, pE` are clear; positions are correctly adjacent for a leftward move.
   - **Effects**: `pA, pC` become clear; `pD, pE` are now occupied by `piece`.

5. **move_l_up(pA, pB, pC, pD, pE, pMid, piece)**: Moves or rotates an L-piece up.
   - **Preconditions**: `piece` occupies `pA`, `pB`, `pC`; `pD`, `pE` are clear; positions are correctly adjacent for an upward move.
   - **Effects**: `pB`, `pC` become clear; `pD`, `pE` are now occupied by `piece`.
6. **move_l_down(pA, pB, pC, pD, pE, piece)**: Moves or rotates an L-piece down.
   - **Preconditions**: `piece` occupies `pA`, `pB`, `pC`; `pD`, `pE` are clear; positions are correctly adjacent for a downward move.
   - **Effects**: `pA`, `pC` become clear; `pD`, `pE` are now occupied by `piece`.

### C.5 FLOOR TILES DOMAIN

The Floor Tiles domain involves robots painting a grid of tiles. Each robot can hold one color at a time and moves between adjacent tiles. A robot can paint an adjacent tile (above or below) with its current color, provided the tile is clear. Once a tile is painted, it cannot be occupied.

ACTIONS

1. **change-color(robot, color-from, color-to)**: A robot changes its held paint color.
   - **Preconditions**: The `robot` is holding `color-from`; `color-to` is an available color.
   - **Effects**: The `robot` is now holding `color-to`.
2. **paint-up(robot, tile-paint, tile-robot, color)**: A robot paints the tile above its current position.
   - **Preconditions**: `robot` is at `tile-robot`; `tile-paint` is directly above `tile-robot`; `tile-paint` is clear; `robot` is holding `color`.
   - **Effects**: `tile-paint` is now painted with `color` and is no longer clear.
3. **paint-down(robot, tile-paint, tile-robot, color)**: A robot paints the tile below its current position.
   - **Preconditions**: `robot` is at `tile-robot`; `tile-paint` is directly below `tile-robot`; `tile-paint` is clear; `robot` is holding `color`.
   - **Effects**: `tile-paint` is now painted with `color` and is no longer clear.
4. **up(robot, tile-from, tile-to)**: A robot moves one tile up.
   - **Preconditions**: `robot` is at `tile-from`; `tile-to` is directly above `tile-from`; `tile-to` is clear.
   - **Effects**: `robot` is now at `tile-to`; `tile-from` becomes clear; `tile-to` is no longer clear.
5. **down(robot, tile-from, tile-to)**: A robot moves one tile down.
   - **Preconditions**: `robot` is at `tile-from`; `tile-to` is directly below `tile-from`; `tile-to` is clear.
   - **Effects**: `robot` is now at `tile-to`; `tile-from` becomes clear; `tile-to` is no longer clear.
6. **right(robot, tile-from, tile-to)**: A robot moves one tile right.
   - **Preconditions**: `robot` is at `tile-from`; `tile-to` is directly to the right of `tile-from`; `tile-to` is clear.
   - **Effects**: `robot` is now at `tile-to`; `tile-from` becomes clear; `tile-to` is no longer clear.
7. **left(robot, tile-from, tile-to)**: A robot moves one tile left.
   - **Preconditions**: `robot` is at `tile-from`; `tile-to` is directly to the left of `tile-from`; `tile-to` is clear.
   - **Effects**: `robot` is now at `tile-to`; `tile-from` becomes clear; `tile-to` is no longer clear.

## D OVERVIEW OF PROMPTS

In this section, we provide a mapping between the main inference functions and their associated prompts, clarifying how the model is prompted at each stage of the search and diagrammatic reasoning pipeline. For each function, we specify: (1) the function name, (2) its core functionality, (3) where it is called in the search process, (4) the key points of its prompt, and (5) its input parameters. This overview enables readers to understand how the model is guided through each step of multimodal reasoning.

| Function Name | Functionality | Where Called | Key Prompt Points | Input Parameters (Description) |
|---|---|---|---|---|
| `next _action` | Selects next best action, generates reasoning, action, and new state description. | During child state generation (generate_child_states). | - Presents problem, initial, current, and goal state (text and images)
- Lists possible actions
- Requests reasoning, action, and new state in code blocks
- Handles uniqueness and previous errors if present | - Problem description (text)
- Possible actions (list)
- Initial, current, goal state objects (text, diagrams, schemas)
- Model name, temperature
- Chosen actions
- Previous attempt/error (optional) |
| `generate _diagram _schema` | Generates diagram schema for a child state after an action. | During child state diagram generation (generate_diagrams). | - Presents problem, initial, current, and new state (text, diagrams, schemas)
- Requests one statement per object (position, size, status, identifier)
- Enforces object count consistency
- Handles previous attempt/error if present | - Problem description (text)
- Initial state object (with schema)
- Child state object (with action, parent, state description)
- Model name, temperature
- Previous attempt/error (optional) |
| `test _diagram _schema` | Verifies correctness of a generated diagram schema for a child state. | After schema generation, before code generation (generate_diagrams). | - Presents problem, initial, current, and child state (text, diagrams, schemas)
- Validates object count, affected object status, and consistency
- Requires yes/no answer and error summary | - Problem description (text)
- Initial state object (with schema)
- Child state object (with schema, action, parent)
- Model name |
| `generate _diagram _code` | Generates Matplotlib code to visualize a child state based on its schema. | After schema validation (generate_diagrams). | - Presents problem, initial/current/child state (text, diagrams, schemas)
- Provides example code and reasoning if available
- Requests code to visualize all objects as described in schema
- Enforces clarity, no overlaps, correct labeling, plausibility
- Handles previous attempt/error if present | - Domain name
- Problem description (text)
- Child state object (with schema, action, parent)
- Initial state object (with code, diagram)
- Model name, temperature
- Save path for image
- Previous attempt/error (optional) |

| Function Name | Functionality | Where Called | Key Prompt Points | Input Parameters (Description) |
|---|---|---|---|---|
| `test _diagram` | Verifies correctness and clarity of a generated diagram image for a child state. | After code execution, before accepting diagram (generate_diagrams). | - Presents problem, current/child state (text, diagrams, schemas) 
 - Validates object presence, status, labeling, readability, plausibility 
 - Requires yes/no answer and error summary | - Problem description (text) 
 - Child state object (with schema, diagram, parent) 
 - Domain name 
 - Model name |
| `check _action _validity` | Checks if a proposed action and resulting state are valid (local verification). | After diagram generation, before path/global verification (generate_child_states). | - Presents problem, initial/current/goal state (text, diagrams, schemas) 
 - Lists action path, action taken, new state description, and diagram 
 - Checks preconditions, effects, diagram accuracy, reasoning 
 - Requires yes/no answer and error summary | - Problem description (text) 
 - Current state object (with schema, diagram) 
 - Action taken (string) 
 - New state description (text) 
 - New state object (with schema, diagram) 
 - Goal state object 
 - Possible actions (list) 
 - Initial state object 
 - Model name |
| `check _action _path` | Checks if the entire action path from initial to current state is valid (global verification). | After local verification, before accepting child state (generate_child_states). | - Presents problem, initial/current/goal state (text, diagrams, schemas) 
 - Lists full action path and parent state 
 - Checks preconditions, effects, diagram accuracy for the whole path 
 - Requires yes/no answer and reasoning | - Problem description (text) 
 - Initial state object (with schema, diagram) 
 - Current state object (with schema, diagram, parent) 
 - Goal state object 
 - Actions (list) 
 - Possible actions (list) 
 - Model name |
| `rank _states` | Ranks candidate states at a given depth by proximity to the goal state (for beam search). | After child state generation, before pruning (beam_search). | - Presents problem, goal state, and all candidate states (text, diagrams, action paths) 
 - Requests ranking based on number of goal constraints satisfied 
 - Requires ranking code block and reasoning | - List of state objects (with id, description, diagram, action path) 
 - Problem description (text) 
 - Goal state object 
 - Model name |

| Function Name | Functionality | Where Called | Key Prompt Points | Input Parameters (Description) |
|---|---|---|---|---|
| `ini_g _diagrams` | Generates and verifies diagrams for initial and goal states. | At the start of search, during problem setup (setup). | - Presents problem, initial/goal state (text)
- Requests diagram schema and code for initial/goal state
- Enforces object coverage, status, and visual consistency
- Verifies schema and diagram correctness
- Handles retries and error feedback | - Problem name, domain name
- Problem description (text)
- Initial state object (with text)
- Goal state object (with text)
- Model name, temperature
- Parameters (e.g., max attempts)
- Error message, previous attempt (optional) |
| `generate _diagram _schema _ini_g` | Generates diagram schema for initial or goal state. | Within ini_g_diagrams (problem setup). | - Presents problem, initial and goal state descriptions
- Requests one statement per object (position, size, status, identifier)
- Handles previous attempt/error if present | - Problem description (text)
- Initial state object (with text)
- Goal state object (with text)
- Model name, temperature
- Domain name
- Goal flag (bool)
- Previous attempt/error (optional) |
| `test _diagram _schema _ini_g` | Verifies correctness of initial/goal state diagram schema. | Within ini_g_diagrams (after schema generation). | - Presents problem, state description, and schema
- Validates object coverage and status
- Requires yes/no answer and error summary | - Problem description (text)
- Initial state object (with schema)
- Goal state object (with schema)
- Model name
- Goal flag (bool) |
| `generate _diagram _code _ini_g` | Generates Matplotlib code for initial or goal state visualization. | Within ini_g_diagrams (after schema validation). | - Presents problem, state description, and schema
- Provides code and reasoning
- Requests code to visualize all objects as described in schema
- Enforces clarity, labeling, plausibility
- Handles previous attempt/error if present | - Domain name
- Problem description (text)
- Initial state object (with schema)
- Goal state object (with schema)
- Model name, temperature
- Save path for image
- Goal flag (bool)
- Previous attempt/error (optional) |
| `test _diagram _ini_g` | Verifies correctness and clarity of initial/goal state diagram image. | Within ini_g_diagrams (after code execution). | - Presents problem, state description, and diagram image
- Validates object presence, status, labeling, consistency, plausibility
- Requires yes/no answer and error summary | - Problem description (text)
- Initial state object (with schema, diagram)
- Goal state object (with schema, diagram)
- Domain name
- Model name
- Goal flag (bool) |

| Function Name | Functionality | Where Called | Key Prompt Points | Input Parameters (Description) |
|---|---|---|---|---|
| `is_unique_action` | Checks if a proposed action leads to a unique child state from the current state. | During child state generation in generate_child_states. | - Presents problem, current state, action taken, and new state description
- Lists previously explored actions and resulting states
- Asks if the new action/state is unique
- Requires yes/no answer and explanation | - Problem description (text)
- Current state object (with child states)
- Action taken (string)
- New state description (text)
- Model name |
| `check_goal_state` | Checks if the current state satisfies all goal state constraints. | At each node expansion in beam_search. | - Presents problem, initial, current, and goal state (text, diagrams)
- Lists action path
- Asks if current state matches all goal constraints
- Requires yes/no answer and step-by-step reasoning | - Problem description (text)
- Current state object (with diagram)
- Goal state object (with diagram)
- Initial state object (with diagram)
- Model name |

This mapping clarifies the modular structure of the inference pipeline and the precise role of each prompt in guiding the model's multimodal reasoning. Each function is responsible for a distinct aspect of the search or verification process, and the prompts are carefully designed to enforce correctness, clarity, and consistency at every stage. The table above can be used as a reference for understanding or extending the codebase, as well as for reproducing or adapting the prompting strategy to new domains or tasks.

# E   CODE STRUCTURE AND ORGANIZATION

The codebase is organized to support multimodal planning and diagrammatic reasoning across multiple domains. Each domain (e.g., `blocksworld`, `barman`, `elevator`, `parking`, `tetris`, `tiles`) is self-contained, with a consistent folder structure and supporting scripts. Below, we describe the main components and their roles.

## E.1   TOP-LEVEL STRUCTURE

- **Domain Folders:** Each domain (e.g., `blocksworld`, `barman`, etc.) is a top-level folder containing all files and subfolders needed to run our method on the instances of that domain.
- **Shared Scripts:** At the root, scripts such as `search.py`, `inference.py`, and diagram generation scripts are provided for general use across domains.
- **Utilities:** Files like `requirements.txt` and `Readme.md` provide environment setup and documentation.

## E.2   KEY SCRIPTS AND THEIR ROLES

- **`search.py`**: The main search and script. Implements the graph of thought algorithm with beam search and backtracking, manages state expansion, diagram generation, and verification. For each problem instance, it creates a subfolder (e.g., `blocksworld_instance_1/`) and stores all intermediate and final results, including state diagrams, code, and logs.
- **`inference.py`**: Contains all functionalities facilitated by LMMs and prompting logic, including functions for generating and verifying actions, diagram schemas, code, and state rankings. This script is the interface between the search process and the language model.
- **`visual_thinking.py`**: Main pipeline script. Orchestrates the batch processing of multiple problem instances for a given domain. For each instance, it sets up the directory structure, runs our diagrammatic search, and validates the resulting plan using VAL. It also manages the output and validation logs.

- **initial_diagram_schema.py**, initial_diagram_code.py, initial_goal_diagram_code.py: Scripts for generating and storing the first initial and goal conceptual diagrams for the domain and their schemas. Outputs are stored in the initial_conceptual_diagram/ folder.
- **PDDL_tranlation.py** (per domain): Handles translation between natural language and PDDL for that domain, including prompt templates and in context examples.

## E.3   INSTANCE AND STATE FOLDER STRUCTURE

For each problem instance (e.g., blocksworld_instance_1/), the following structure is created during search:

- state_X/: For each explored state, a folder is created containing:
  – diagram.png: The generated diagram for the state.
  – diagram_code.py: The code used to generate the diagram.
  – diagram_schema.txt: The schema describing the diagram.
  – info.txt: Metadata about the state, including parent, action taken, and reasoning.
  – attempts/: Subfolders for storing all attempts at generating child states and diagrams, including error logs.
- goal_state/: If a goal is found, this folder contains the final state information and a copy of all diagrams along the solution path.
- ranking/: Stores state ranking information at each search depth.
- output.txt, plan.pddl, val_output.txt: Output logs, the generated plan, and validation results.

## E.4   DOMAIN EXAMPLES

The repository attached in the supplementary material includes several PDDL domains (blocksworld, barman, elevator, parking, tetris, tiles), each with a complete set of PDDL files, instance generators, randomly generated instances, and translations of the domain and a random state and initial conceptual diagrams generated using pipeline. For blocksworld, we also provide full examples of solved problems, including all intermediate diagrams and logs, to facilitate reproducibility and intuitive understanding of the pipeline. To view the sequence of diagrams produced for a successfully solved instance, refer to the goal_state folder within each instance's directory. This folder contains the chain of diagrams generated by the pipeline from the initial state to the goal state for that instance.

This modular structure allows for easy extension to new domains and facilitates reproducibility, inspection, and further research.

