# OpenReview forum: "Visualizing Thought: Conceptual Diagrams Enable Robust Planning in LMMs"
_ICLR.cc/2026/Conference — Submitted to ICLR 2026_

### Official Review · Reviewer_YThF · 2025-10-25

**Soundness:** 4
**Presentation:** 3
**Contribution:** 4
**Rating:** 6
**Confidence:** 4

**Summary:**

This paper introduces "Visual Thinking," a framework that significantly enhances the performance of Large Multimodal Models (LMMs) on combinatorial planning tasks by enabling them to generate and reason with conceptual diagrams autonomously. The method integrates beam search and backtracking mechanisms within a Graph-of-Thought inference framework, requiring no manually drawn diagrams or domain-specific templates. Evaluations across multiple PDDL planning domains demonstrate that the approach substantially outperforms both pure-text reasoning models and search-based baselines. The paper also contributes a new challenging benchmark for long-horizon planning.

**Strengths:**

1. **Originality:** The paper is highly original, being the first to introduce autonomously generated conceptual diagrams as a reasoning medium for LMMs, combined with a graph-based reasoning framework and search optimization strategies. Compared to existing methods that rely on human-drawn diagrams or single-step visual aids, this approach is more generalizable and scalable.

2. **Quality:** The experimental design is rigorous, encompassing multiple planning domains, including a newly proposed challenging benchmark. The ablation studies thoroughly validate the contribution of each component.
Significance: This research not only technically advances the planning capabilities of LMMs but also opens up new directions for multimodal reasoning.

**Weaknesses:**

1. **Substantial Computational Overhead:** Although the paper claims the method is more efficient than text-only search, the generation and verification of diagrams still introduce significant computational costs.

2. **Domain Limitations:** The method is currently limited to combinatorial planning problems expressible in PDDL. It has not been tested on more open-ended, unstructured reasoning tasks. The high error rate in domains with high branching factors like Tetris also indicates room for improvement in handling complex, parameterized actions.

**Questions:**

* Q1: The paper does not discuss how consistency in diagram generation affects long-horizon reasoning. Were there instances where inconsistencies in diagram style across steps led to reasoning breakdowns? Is there any mechanism in place to ensure coherence between consecutive diagrams in a multi-step sequence?

* Q2: Is the method applicable to tasks not described in PDDL, such as everyday tasks specified by natural language instructions? Are there plans to extend the evaluation to broader reasoning benchmarks to demonstrate generalizability beyond formal planning domains?

* Q3: Regarding the dramatic performance drop when replacing rendered diagrams with their underlying code, was a deeper analysis conducted on the specific bottlenecks the model faced when processing the code? Were alternative, simplified code representations or intermediate visual formats explored to mitigate this issue?

---

> ### Author Response · Authors · 2025-11-21
>
> Thank you for the thoughtful review. We are glad you found the paper original. Below, we address your questions and concerns.
>
> > Substantial Computational Overhead: … the generation and verification of diagrams still introduce significant computational costs.
>
> To isolate the cost of adding diagrams, the most direct comparison is between our full method (GPT-4o + Visual Thinking) and the Optimized GoT baseline, which uses the same text-based graph search with beam search and backtracking but without any diagram components and inferences. Our analysis showed that incorporating diagrams into our pipeline added an average of 213 seconds in latency, representing a 24.4% increase (872s → 1085s), and 0.71 USD in cost, representing a 27.7% increase (2.49 USD → 3.18 USD) per problem compared to the text-only Optimized GoT baseline. This overhead delivers a 30% improvement in accuracy, which is a comparatively favorable trade-off.
>
> Importantly, our method remains significantly more efficient than search-based baselines proposed in prior research due to the use of beam search. On average, it was 31% faster and 36% cheaper than the standard GoT baseline across all domains, and 46% faster and 52% cheaper than RAP+GPT-4o, while also being substantially more accurate (outperforming GoT by 43% and RAP by 22%).
>
> It’s also important to note that our diagrams are lightweight Matplotlib renderings of basic shapes/labels, not photorealistic images. As multimodal model cost for visual inputs scales with image resolution, these conceptual diagrams are far more cost-effective than photos or photorealistic images, preserving efficiency while providing substantial reasoning gains.
>
> > The method is currently limited to combinatorial planning problems expressible in PDDL. ...  Q2: Is the method applicable to tasks not described in PDDL, such as everyday tasks specified by natural language instructions?
>
> We agree that demonstrating generalization to real-world planning scenarios is crucial. As described in Section 3 of the paper, our method takes only a natural-language description of the task as input and automatically generates both textual and diagrammatic representations through a search-based inference process—no human initialization or domain-specific syntax is required.
>
> Importantly, our method does not operate directly on PDDL syntax. PDDL is used solely as a source of diverse planning domains and as a validation interface for checking the correctness of generated plans. Because all reasoning is performed in natural language, our approach generalizes directly to text-based real-world environments.
>
> To support this claim and demonstrate the generalization of the method to real world scenarios, we added new experiments on the **NATURAL PLAN** dataset (Zheng et al., DeepMind 2024), which includes Meeting Planning and Calendar Scheduling tasks with API-based text outputs (Flights, Maps, Calendar) integrated in the problem description. We evaluated 100 instances per domain using the dataset’s provided evaluation pipeline. Results are summarized below:
>
> | Model | Meeting Planning | Calendar Scheduling |
> |:------|:----------------:|:------------------:|
> | GPT-3.5 | 19.1 % | 19.9 % |
> | GPT-4  | 47.0 % | 41.2 % |
> | GPT-4o | 45.2 % | 43.7 % |
> | Gemini 1.5 Flash | 23.9 % | 34.3 % |
> | Gemini 1.5 Pro | 39.1 % | 48.9 % |
> | **Visual Thinking (GPT-4o)** | **83 %** | **91 %** |
>
> All baseline results are taken from the NATURAL PLAN paper. Visual Thinking achieves 83 % and 91 % exact-match accuracy respectively—far surpassing all baselines without any change to the original prompt or pipeline. This demonstrates robust generalization to complex natural-language planning problems. Example generated diagrams for these domains appear in Appendix A.

---

> ### Author Response · Authors · 2025-11-21
>
> >Were there instances where inconsistencies in diagram style across steps led to reasoning breakdowns? Is there any mechanism in place to ensure coherence between consecutive diagrams in a multi-step sequence?
>
> We acknowledge the importance of maintaining consistency of diagrams during multi-step reasoning. Our framework enforces style and semantic consistency between different states of a single instance and also across different instances of a domain using the following mechanisms:
>
> 1. **Domain-level template caching.**
>    During the (automatic) setup of a new domain, we sample multiple diagram schemas and codes for a random instance of the domain, render and verify the diagrams, rank them by representational clarity, and cache the Matplotlib code of the top ranked diagram. This code is stored and later used in the prompts as the reference diagram for generating the first diagram of instances in this domain (Fig. 2).
> 2. **Conditioning on domain reference code.**
>    Initial and goal diagrams for each instance are generated conditioned on this cached reference code (Fig. 1, Step 2), ensuring fixed mapping between entities and their interactions and shapes, colors, legends, and spatial layouts across all instances.
> 3. **Parent code conditioning for child states.**
>    For every expansion, child-state diagram code is generated conditioned on the diagram schema and the codes for the initial and parent states, ensuring style and conceptual mapping  consistency within each search path (Sec. 3).
> 4. **Verification guardrails.**
>   The local self check at the end of the child generation pipeline verifies consistency between diagram and diagram schema of the parent and child states and ensures consistent and accurate visual representation of colors, positions, and statuses between the two states, considering the action taken, preventing drift between consecutive diagrams (Fig. 3; Sec. 3).
>
> Consequently, during diagram code generation for every new state at least one reference diagram code is included in the prompt, guiding the model to maintain both stylistic and conceptual consistency. Empirically, we did not observe style inconsistencies causing failures. As shown in Figure 4 and run folders in the supplementary material, legends, color semantics, and glyphs remain stable across action sequences and within each domain.
>
> > Regarding the dramatic performance drop when replacing rendered diagrams with their underlying code.. were alternative, simplified code representations or intermediate visual formats explored to mitigate this issue?
>
> Our various ablation experiments isolate the effect of different intermediate state representations on the downstream planning capability of the model (Table 4).
>
> - Full method (diagrams): 90.2% on Blocksworld(simple).
>
>
> - No Diagram Schema: 72% (code generated directly from text).
>
>
> - No Diagram (Optimized GoT, text-only): 58%.
>
>
> - No Code Execution (provide code instead of rendered diagram): 24%.
>
>
> To further address your questions and analyze the effect of different representations on planning performance, we evaluated another ablation: **Diagram Schema Only** (no diagram code generation or visuals) where the state representation shown to the model during planning includes a textual description and the diagram encoding of the state,  an intermediate structured textual representation encoding the relative relationship between objects. In our full pipeline we condition code generation on this representation to ensure all interactions are accurately visualized in a spatial layout. This method  achieved **66% correct**, 28% incorrect, 6% incomplete on Blocksworld(simple)—better than text‑only (58%) but below the full diagrammatic pipeline (90.2%). This supports the hypothesis that structured relational text representations help planning performance, while confirming that rendered diagrams help the most.
>
> Thus, throughout our ablations we evaluate many representations of data and show that:
>
> _conceptual diagrams > structured relational encoding in text > free text representations > code_
>
> The sharp performance drop (from 90.2% to 24%) when using raw Matplotlib code—rather than rendered diagrams—reflects how poorly code structure represents the relational information. Raw code has verbose syntax (variable names, loops, style commands) which obscures core information. The model must search irrelevant tokens to extract object status, making reasoning harder.
>
> For an intuitive example, please consider the following example. In a simple game like tic-tac-toe, given an image of the board, we can instantly grasp the state and available moves. Given a block of code that draws the board, even with all information present, one must parse every line, decipher variable assignments, and reconstruct the spatial layout mentally. This is much harder, slower, and error-prone—even compared to a simple list of cell states in text.

---

> > ### Author Response · Authors · 2025-11-21
> >
> > We hope this response addresses your questions and concerns and highlights how Visual Thinking offers a general and efficient way for multimodal reasoning in LMMs that aligns with human cognition. Thank you again for the constructive feedback.

---

### Official Review · Reviewer_Uw6N · 2025-10-29

**Soundness:** 2
**Presentation:** 2
**Contribution:** 2
**Rating:** 4
**Confidence:** 4

**Summary:**

In this paper, the authors propose to (using prompt primarily) conduct visual thinking for VLLMs. Specifically, the method involves textual and diagrammatic reasoning within an optimized Graph-of-Thought inference framework, together with beam search and backtracking. The authors primarily evaluate their prompting methods with LLMs from the recent GPT families (GPT-4o, o1-mini, and o1-preview) on several datasets. The result shows the promise of feeding both the diagram and the text in the reasoning process.

**Strengths:**

- The authors have demonstrated, at least for the GPT set of models, their method of prompting can enhance the model's reasoning on several datasets (which I would not call real-world, as from my understanding, they are simulated, including the ones proposed by the authors).
- The proposed method has significantly improves the GPT models' performance on the datasets evaluated in this paper.

**Weaknesses:**

- How does the base model selection affect the outcome? For instance, for open-source VLLMs such as models from QWen family and other VLLM models?
- Are there any real world datasets that your method can help with? For instance, [1] tested their prompting method on several real-world planning tasks, [2] tested their methods on table-specific applications. I think including diverse datasets such as complicated real-world planning tasks, higher order theory of mind reasoning would demonstrate the value of the prompting methods.
- One of the core arguments of this paper is to augment the text reasoning trace with the visual parts. This reminds of the recent discussion on reasoning with text versus images [3, 4] and many others. It would be nice for the authors to include these relevant works in the related work section.

----
### References

[1] Sun, Zhenjie, et al. "Table as Thought: Exploring Structured Thoughts in LLM Reasoning." arXiv preprint arXiv:2501.02152 (2025).

[2] Wang, Zilong, et al. "Chain-of-table: Evolving tables in the reasoning chain for table understanding." arXiv preprint arXiv:2401.04398 (2024).

[3] Wei, Haoran, Yaofeng Sun, and Yukun Li. "DeepSeek-OCR: Contexts Optical Compression." arXiv preprint arXiv:2510.18234 (2025).

[4] Deng, Naihao, et al. "Tables as texts or images: Evaluating the table reasoning ability of llms and mllms." arXiv preprint arXiv:2402.12424 (2024).

**Questions:**

See weaknesses.

---

> ### Author Response · Authors · 2025-11-21
>
> We appreciate the constructive review and the concrete pointers to related work. Below, we address your questions and concerns.
>
> >How does the base model selection affect the outcome?
>
> Our method is model‑agnostic by design; the gains come from more compact and parallel representation of information in conceptual diagrams and efficient search-based inference, and are not dependent on any particular model backbone. Besides the GPT‑4o results reported in Table 1, in the original version we showed strong transfer on Claude 3.5 Sonnet and Llama 4 models , 54.8% to 98% and 10% to 74% respectively (Sec. 4.1 “Model Generalization”). To fully address your concern, we  have now added evaluation using **Qwen3‑VL** (open‑source VLLM) on Blocksworld (simple) with the same VIsual Thinking prompts/hyper‑parameters used in our main experiment results. The perfromance of this model improved from **28\% using single-inference to 78% using Visual Thinking**. Full set of model generalization results are summarized in Table 4.
>
> > Are there any real world datasets that your method can help with?
>
> We agree that demonstrating generalization to real-world planning scenarios is crucial. As described in Section 3 of the paper, our method takes only a natural-language description of the task as input and automatically generates both textual and diagrammatic representations through a search-based inference process—no human initialization or domain-specific syntax is required.
>
> Importantly, our method does not operate directly on PDDL syntax. PDDL is used solely as a source of diverse planning domains and as a validation interface for checking the correctness of generated plans. Because all reasoning is performed in natural language, our approach generalizes directly to text-based real-world environments.
>
> To support this claim and demonstrate the generalization of the method to real world scenarios, we added new experiments on the **NATURAL PLAN** dataset (Zheng et al., DeepMind 2024), which includes Meeting Planning and Calendar Scheduling tasks with API-based text outputs (Flights, Maps, Calendar) integrated in the problem description. We evaluated 100 instances per domain using the dataset’s provided evaluation pipeline. Results are summarized below:
>
> | Model | Meeting Planning | Calendar Scheduling |
> |:------|:----------------:|:------------------:|
> | GPT-3.5 | 19.1 % | 19.9 % |
> | GPT-4  | 47.0 % | 41.2 % |
> | GPT-4o | 45.2 % | 43.7 % |
> | Gemini 1.5 Flash | 23.9 % | 34.3 % |
> | Gemini 1.5 Pro | 39.1 % | 48.9 % |
> | **Visual Thinking (GPT-4o)** | **83 %** | **91 %** |
>
> All baseline results are taken from the NATURAL PLAN paper. Visual Thinking achieves 83 % and 91 % exact-match accuracy respectively—far surpassing all baselines without any change to the original prompt or pipeline. This demonstrates robust generalization to complex natural-language planning problems. Example generated diagrams for these domains appear in Appendix A.
>
>
> >  It would be nice for the authors to include these relevant works in the related work section.
>
> Thank you for the pointers to the relevant work. In revision, we expanded Section 2 (Related Work) to include Table as Thought [1], Chain-of-Table [2], Tables as Texts or Images [4], and DeepSeek-OCR [3]. These additions clarify that our approach shares the goal of leveraging structured intermediates but differs by maintaining a multimodal state node— i.e. we maintain text + diagram schema + rendered conceptual diagram for each state. We also discuss how Tables as Texts or Images supports our finding that format, not only content, effect reasoning and planning performance.
>
> Additionally, to further respond to this feedback and compare our visual representation to structured textual representations, we added a new **structured-text ablation, the Diagram Schema-only run** in Table 5. The diagram schema is a compact relational encoding of objects, adjacency/containment, and statuses generated in our pipeline on which we condition each state’s diagram code generation (Section 3 and Figure 3) (See Appendix B for example diagram schemas).
>
> In the “only diagram schema” ablation the textual representation of each state was augmented with the diagram schema during planning and no diagram code was generated. On Blocksworld (simple), this variant achieves 66 % accuracy, higher than text-only search (58 %) but below the full multimodal method (90.2 %), triangulating the contribution of both structured textual representations and conceptual diagrams, while also highlighting the superiority of conceptual diagrams as a medium for parallel and efficient representation of information during reasoning and planning.
>
> --
>
> We hope this response addresses your questions and concerns and highlights how Visual Thinking offers a general and efficient way for multimodal reasoning in LMMs that aligns with human cognition. Thank you again for the constructive feedback.

---

> ### Comment · Reviewer_Uw6N · 2025-11-28
> **Response**
>
> I thank the authors for their response. I like your experiments on the real world datasets. The response has addressed all of my concerns carefully.
>
> Since the authors have already conducted experiments on some natural language tasks. I would be happy to see the authors to conduct a more comprehensive experiments. There are other tasks such as [1] and many other tasks such as math or coding ones. I hope to see the authors to demonstrate some promising results on this.
>
> [1] https://arxiv.org/abs/2402.01622?

---

### Official Review · Reviewer_Uj4f · 2025-10-30

**Soundness:** 3
**Presentation:** 3
**Contribution:** 3
**Rating:** 6
**Confidence:** 3

**Summary:**

Visual Thinking is a framework that enhances large multimodal models by integrating textual and diagrammatic reasoning through a Graph-of-Thought approach. It employs self-generated diagrams with beam search and backtracking to boost complex planning performance.

**Strengths:**

The author integrates graphical representations to allow the model to intuitively grasp reasoning structures via visual information, thus enhancing task performance.

**Weaknesses:**

It is uncertain whether any task can produce a concept, which would limit its ability to generalize.

**Questions:**

How are PDDL and graphs combined in a MLLM? Please explain.

Could you explain how graphs are dynamically generated?

---

> ### Author Response · Authors · 2025-11-21
>
> Thank you for reviewing our work. Below, we address your questions and concerns.
>
> > limit its [method's] ability to generalize.
>
> We agree that demonstrating generalization to real-world planning scenarios is crucial. As described in Section 3 of the paper, our method takes only a natural-language description of the task as input and automatically generates both textual and diagrammatic representations through a search-based inference process—no human initialization or domain-specific syntax is required.
>
> Importantly, our method does not operate directly on PDDL syntax. PDDL is used solely as a source of diverse planning domains and as a validation interface for checking the correctness of generated plans. Because all reasoning is performed in natural language, our approach generalizes directly to text-based real-world environments.
>
> To support this claim and demonstrate the generalization of the method to real world scenarios, we added new experiments on the NATURAL PLAN dataset (Zheng et al., DeepMind 2024), which includes Meeting Planning and Calendar Scheduling tasks with API-based text outputs (Flights, Maps, Calendar) integrated in the problem description. We evaluated 100 instances per domain using the dataset’s provided evaluation pipeline. Results are summarized below:
>
> | Model | Meeting Planning | Calendar Scheduling |
> |:------|:----------------:|:------------------:|
> | GPT-3.5 | 19.1 % | 19.9 % |
> | GPT-4  | 47.0 % | 41.2 % |
> | GPT-4o | 45.2 % | 43.7 % |
> | Gemini 1.5 Flash | 23.9 % | 34.3 % |
> | Gemini 1.5 Pro | 39.1 % | 48.9 % |
> | **Visual Thinking (GPT-4o)** | **83 %** | **91 %** |
>
>
> All baseline results are taken from the NATURAL PLAN paper. Visual Thinking achieves 83 % and 91 % exact-match accuracy respectively—far surpassing all baselines without any change to the original prompt or pipeline. This demonstrates robust generalization to complex natural-language planning problems. Example generated diagrams for these domains appear in Appendix A.
>
> > How are PDDL and graphs combined in an MLLM?
>
>
> Our method does not operate on PDDL input. PDDL is used solely to define diverse planning environments; all experiments are performed on textual representations of the problems derived from those domains.
>  Each domain and instance (including initial and goal states) is translated into natural language using a few-shot prompt following the procedure in Section 4 (“Translating PDDL to Natural Language and Back”). Consequently, all search-based baselines and our method receive the same text-only task descriptions.
>
> Each problem instance therefore consists of: (i) a natural-language description of the objects, admissible actions, and constraints; and (ii) a natural-language specification of the initial and goal states.
>
>
> During inference, the model is provided with a state and asked to predict the next valid action that advances toward the goal from that state. The output—comprising the proposed action and the resulting successor state—forms a node in an evolving graph of states.
>
>  In text-only baselines such as Graph-of-Thought (GoT) and Optimized GoT, each node contains only a textual description of the state. In contrast, Visual Thinking augments each node with (i) a Diagram Schema, a structured textual encoding of object relationships, and (ii) a rendered conceptual diagram. This compact multimodal representation allows the model to reason over spatial and relational structure directly and in parallel, rather than relying only on text and the serialized representation of the interactions.

---

> ### Author Response · Authors · 2025-11-21
>
> >Could you explain how graphs are dynamically generated?
>
> Definitely, we walk through the search graph expansion below; this process is also outlined in Figure 1 and in Section 3 of the manuscript. The process proceeds as follows:
>
> Domain Initialization: The model autonomously generates and ranks multiple diagram schemas and executable diagram codes for a representative instance (Fig. 2). The top-ranked code becomes the reference for subsequent instance-specific diagrams.
>
>
> State Expansion: Beginning from the initial state, the model iteratively proposes up to four actions per node (branching factor = 4). Each action produces a child state consisting of a textual description, a validated diagram schema, and a rendered diagram (Fig. 3).
>
>
> Verification and Search Control: Every generated child state undergoes verification to ensure the selected action is admissible and the state is updated correctly based on the chosen transition. Valid states are added as new nodes. Beam search ranks these candidates by proximity to the goal, considering only the top-k nodes at each depth for further expansion..
>
>
> Termination: Before expanding each state the model checks if the given state matches the goal state. The search concludes the goal state is reached or the computational budget is exhausted (when a maximum number of states is generated or a maximum depth in the search tree is reached). The resulting multimodal graph encodes the full plan trajectory; examples of complete state sequences ina n output plan across domains are provided in Figure 4.
>
>
> An Example: In the Elevator domain, each state describes the current positions of elevators and passengers. Admissible actions include move-up, move-down, board, and leave. The model begins with the initial configuration and sequentially expands possible next actions, each forming a new node connected by the chosen transition. As the search proceeds, the method maintains a structured, multimodal graph representing partial plans from the initial state, with people and elevators being at various floors, until a path satisfying the goal constraints (the final position of people) is found.
>
>  --
>
> We hope this response addresses your questions and concerns and highlights how Visual Thinking offers a general and efficient way for multimodal reasoning in LMMs that aligns with human cognition. Thank you again for reviewing our work.

---

### Official Review · Reviewer_Kbjc · 2025-11-01

**Soundness:** 3
**Presentation:** 2
**Contribution:** 2
**Rating:** 4
**Confidence:** 3

**Summary:**

This paper introduces diagram-based visual thinking, a framework that enables large multimodal models (LMMs) to reason through multiple self-generated diagrams. Instead of relying solely on textual descriptions, it represents intermediate reasoning states visually. The approach requires only a natural language description of the task—without any additional human input—and consistently outperforms text-based baselines such as GoT, optimized GoT, and the o1-series models.

**Strengths:**

(1) The paper presents a detailed efficiency analysis of both generation cost and time compared with baseline methods, which is an important contribution. It also includes thorough ablation studies and analyses of the diagram generation process.

(2) The proposed method is evaluated across multiple game environments, with clear and well-structured methodological descriptions.

**Weaknesses:**

(1) The baselines using GoT and optimized GoT appear unreasonably weak. Could you provide more details about their outputs? If the task states are clearly described, GoT should achieve relatively strong performance on these tasks. It would be helpful to include some basic case studies and an analysis of the error types to clarify this gap.

(2) While complex visual reasoning might be necessary for robotics tasks, the game-like environments used here may not require such intricate visual representations. Similar to environments like TextWorld or AlfWorld, the key lies in the language used to describe the relationships between objectives. A more cost-effective baseline could focus on describing these relationships rather than the absolute states of each object. For instance, in Figure 3, instead of generating diagram code, one could produce text captions emphasizing relative relationships (e.g., “Object A is held by the hand above; below, from left to right, are A, B, and C”). You could include baselines that generate such captions using different prompts—one focusing on absolute states and another on relational descriptions.

(3) The generalization ability of the proposed method remains questionable. For real-world multimodal planning tasks, such as embodied reasoning, abstracting complex environments into diagrams is non-trivial. Additional experiments on more challenging and realistic scenarios would strengthen the paper’s claims about generalization. It is ok to provide just one or two examples to convince me it is working under more complicated tasks.

**Questions:**

How are the individual data point tasks constructed for each of the five additional IPC domains—Floor Tiles, Parking, Tetris, Elevator, and Barman? Do you add special difficulty for them? Could you provide more details about their design beyond the action space and brief task descriptions?

---

> ### Author Response · Authors · 2025-11-21
>
> We thank you for the thoughtful feedback and concrete suggestions. Below we clarify the evaluation setup, provide the requested deeper analysis of GoT/Optimized GoT, discuss the role of “Diagram Schema” in our pipeline as a structured textual relational encoding, present new generalization results on NATURAL PLAN, and detail how we construct instances for the five additional IPC domains.
>
> >The baselines using GoT and optimized GoT appear unreasonably weak. Could you provide more details about their outputs?
>
> All methods (GoT, Optimized GoT, RAP, and our approach) receive the same natural‑language description of each PDDL domain. We translate PDDL domains to text with a fixed five‑shot prompt, the domain natural language description (includes descriptions of possible actions, their preconditions and effects along with description of objects present in each state) (See Appendix C) is generated once and then cached for subsequent use by all evaluated methods,  and translate each instance (initial and goal state) with a fixed one‑shot prompt (Section 4, Translating PDDL to Natural Language and Back). No method is given extra “hints” beyond these textual specifications.
>
> Requested deeper analysis of GoT/Optimized GoT: Below we report additional diagnostics for GoT and Optimized GoT baselines (states expanded, correct solution depth ranges, and failure modes).
>
> ### Table 1: Graph-of-Thought performance analysis (BFS, text-only states)
>
> | Domain               | Correct (%) | Incorrect (%) | Incomplete (%) | Depth avg (correct) | Depth min (correct) | Depth max (correct) | Avg # states |
> |----------------------|-------------|----------------|----------------|----------------------|------------|------------|---------------|
> | Blocksworld-simple   | 50          | 28             | 22             | 6.6                 | 2          | 13         | 59.3          |
> | BlocksWorld-Hard     | 8           | 2              | 90             | 22.0                   | 20         | 26         | 380.2         |
> | Elevator             | 2           | 12             | 86             | 18.0                   | 18         | 18         | 231.5         |
> | Parking              | 14          | 60             | 26             | 10.0                | 4          | 19         | 125.6         |
> | Tetris               | 0           | 70             | 30             | —                    | —          | —          | 141.1         |
> | Floor Tiles          | 0           | 8              | 92             | —                    | —          | —          | 372.3         |
> | Barman               | 0           | 4              | 96             | —                    | —          | —          | 420.2         |
>
> ---
>
> ### Table 2: Optimized GoT (beam search at the frontier; text-only states)
>
> | Domain               | Correct (%) | Incorrect (%) | Incomplete (%) | Depth avg (correct) | Depth min (correct)| Depth max (correct)| Avg # states |
> |----------------------|-------------|----------------|----------------|----------------------|------------|------------|---------------|
> | BlocksWorld (simple) | 58          | 30             | 12             | 7.6                 | 0          | 16         | 43.2          |
> | BlocksWorld-Hard     | 48          | 22             | 30             | 19.2                | 13         | 32         | 161.7         |
> | Elevator             |10           | 50             | 40             |   17.4                  | 16          | 24          | 175.2         |
> | Parking              | 28          | 52             | 20             | 8.6                  | 2          | 20         | 121.4         |
> | Tetris               | 12          | 88             | 0              | 5.8                  | 4          | 11         | 48.3          |
> | Tiles                | 4           | 56             | 40             | 25.4                 | 21         | 30         | 208.6         |
> | Barman               | 0           | 38             | 62             | —                 | —         | —         | 286.0         |

---

> ### Author Response · Authors · 2025-11-21
>
> The base GoT uses breath-first search (BFS) to expand text‑only states; for long‑horizon planning (solution depths up to ~40 by design in our new benchmark) this leads to a combinatorial explosion in the number of states generated, quickly exhausting the computational budget we set for all search-based methods (Section 3 and Table 2). For example, with GoT the search is almost always (for 92% of instances) incomplete on Tiles (avg. 372 states generated), 96% incomplete on Barman (avg. 420 states generated), and 90% incomplete on BlocksWorld‑Hard (avg. 380 states). Optimized GoT uses beam search instead of BFS to expand the search tree, which significantly reduces inefficient expansions and improves accuracy in several domains (e.g., BlocksWorld‑Hard, Tetris), but it is still constrained by a text‑only serialized state representation. In our qualitative error analysis of text‑only search with optimized GoT, two failure modes recur:
> - Do/undo oscillations (e.g., in Parking and Elevator), where the model alternates between a small set of actions without net progress—often when intermediate placements (e.g., staging passengers on non‑destination floors; temporarily double‑parking) are required.
>
>
> - Precondition/object status hallucinations (e.g., Floor Tiles and Tetris), where the model incorrectly hallucinates that the preconditions of an action hold (e.g., whether a tile is clear (i.e., unoccupied or unpainted) to move to) or does not fully update all objects in the state after taking an action.
>
>
> Optimized GoT prunes a text‑only BFS with beam search, but it still encodes each state as a long, linear string—so as objects and constraints grow, the inefficient and serialized representation bottleneck leads to do/undo loops and missed preconditions. Visual Thinking replaces that bottleneck with compact multidimensional (2‑D +colors) conceptual diagrams while using the same beam/backtracking, making relations and statuses on multiple objects  understandable in parallel and keeping the search focused.
>
> >..similar to environments like TextWorld or AlfWorld, the key lies in the language used to describe the relationships between objectives.
>
> We agree that structured textual schemas improve efficiency of representing relational information. Our pipeline explicitly generates an intermediate Diagram Schema—a compact relational encoding of objects, adjacency/containment, and statuses used to generate each state’s diagram code. This mechanism is described in Section 3 and Figure 3, and implemented via dedicated prompts (See Appendix D for prompt overviews and supplementary materials for full prompts) (See Appendix B for concrete examples of diagram encodings for two domains).
> .
> In the ablation section of the paper, we conducted an experiment to directly measure the contribution of these structured relational encoding on the downstream planning performance of the model. The No Diagram Schema (generate code directly from text) in Table 5 shows accuracy drops from 90.2% → 72% when diagram code generation is only conditioned on raw text, emphasizing the rule of these clear structured statements on enabling the model to accurately represent the object interactions.
>
>
> Moreover, to directly address your inquiry about replacing diagrams with a textual relational encoding, we additionally ran a **Diagram Schema‑only (no visual diagram)** variant on Blocksworld (simple) which yielded the following results (updated in Table 5): 66% correct, 28% incorrect, 6% incomplete. This accuracy sits between No Diagram (58%) and the full method (90.2%), improving the rate of incorrect solutions. This aligns with your hypothesis that structured relational summaries help; however, we find the conceptual diagrams to be the most efficient and helpful representation of information for long horizon planning.
>
> We believe this gain comes from representing multi‑object relations in a spatial, parallel format (the 2-D layout and color-coded status) which will be preserved in convolutional attention layers when passing in diagrams as images but lost when the same information is serialized in linear text. This multidimensional representation reduces both hallucinations and do/undo loops, particularly in high-object, high-depth domains like Barman.

---

> ### Author Response · Authors · 2025-11-21
>
> >The generalization ability of the proposed method [to real world scenarios] remains questionable.
>
> We agree that demonstrating generalization to real-world planning scenarios is crucial. As described in Section 3 of the paper, our method takes only a natural-language description of the task as input and automatically generates both textual and diagrammatic representations through a search-based inference process—no human initialization or domain-specific syntax is required.
>
> Importantly, our method does not operate directly on PDDL syntax. PDDL is used solely as a source of diverse planning domains and as a validation interface for checking the correctness of generated plans. Because all reasoning is performed in natural language, our approach generalizes directly to text-based real-world environments.
>
> To support this claim and demonstrate the generalization of the method to real world scenarios, we added new experiments on the NATURAL PLAN dataset (Zheng et al., DeepMind 2024), which includes Meeting Planning and Calendar Scheduling tasks with API-based text outputs (Flights, Maps, Calendar) integrated in the problem description. We evaluated 100 instances per domain using the dataset’s provided evaluation pipeline. Results are summarized below:
>
> | Model | Meeting Planning | Calendar Scheduling |
> |:------|:----------------:|:------------------:|
> | GPT-3.5 | 19.1 % | 19.9 % |
> | GPT-4  | 47.0 % | 41.2 % |
> | GPT-4o | 45.2 % | 43.7 % |
> | Gemini 1.5 Flash | 23.9 % | 34.3 % |
> | Gemini 1.5 Pro | 39.1 % | 48.9 % |
> | **Visual Thinking (GPT-4o)** | **83 %** | **91 %** |
>
>
> All baseline results are taken from the NATURAL PLAN paper. Visual Thinking achieves 83 % and 91 % exact-match accuracy respectively—far surpassing all baselines without any change to the original prompt or pipeline. This demonstrates robust generalization to complex natural-language planning problems. Example generated diagrams for these domains appear in Appendix A.
>
>
> > How are the individual data point tasks constructed for each of the five additional IPC domains?
>
> All five domains are defined following the International Planning Competition (IPC) format using publicly available PDDL instance generators [1].
>
> To maintain consistent difficulty across domains, we matched the maximum possible solution depth (~40) observed in the BlocksWorld-Hard benchmark (Valmeekam et al., 2024 [56]). Instance parameters (e.g., grid size, robot count, or car count) were tuned to yield comparable planning depths. Examples: Floor Tiles: solution depth proportional to # tiles × # robots × movement directions. Parking: constrained by # available curbs x #cars. Tetris: grid size x # tiles.
> Formal action definitions for each domain are listed in Appendix C, including preconditions, effects, and predicate sets. Full problem description for two domains is provided in Appendix B. Full domain instances, generated diagrams, and code for generating instances including hyperparameters and seeds, along with the pipeline’s code, are provided in the supplementary material.
>
> [1] https://ipc02.icaps-conference.org/domains.html
>
> --
>
> We hope this response addresses your questions and concerns and highlights how Visual Thinking offers a general and efficient way for multimodal reasoning in LMMs that aligns with human cognition. Thank you again for the constructive feedback.

---

### Meta-Review · Area_Chair_qLCe · 2026-01-08

**Summary:**

The reviewers acknowledged the novelty of the "generative visual chain-of-thought" concept and the rigorous ablation studies separating the effects of diagram schemas vs. rendered pixels. The rebuttal included additional experiments to address generalization concerns. However, despite the "marginally above" scores from two reviewers, several fundamental limitations prevent this work from meeting the bar for acceptance at this time. The consensus leans towards rejection

**Reviewer Concerns:**

The core critique concerns the scope of applicability. The claim that this is a generalizable reasoning framework for LMMs is not fully supported.

The massive jump in performance suggests that the baseline might be under-optimized, inflating the perceived relative gain of the visual component.

The proposed method is computationally expensive.

**Reviewer Scores:**

The generalization to broader reasoning contexts is unproven, and the efficiency trade-offs are steep. The paper would benefit from a more focused scope or a more robust demonstration of utility in abstract, non-spatial domains.

---

### Decision · Program_Chairs · 2026-01-26

Reject